# Adjustment for Confounding using Pre-Trained Representations

**Rickmer Schulte** [1 2] **David Rügamer** [1 2] **Thomas Nagler** [1 2]

## Abstract

There is growing interest in extending average treatment effect (ATE) estimation to incorporate non-tabular data, such as images and text, which may act as sources of confounding. Neglecting these effects risks biased results and flawed scientific conclusions. However, incorporating non-tabular data necessitates sophisticated feature extractors, often in combination with ideas of transfer learning. In this work, we investigate how latent features from pre-trained neural networks can be leveraged to adjust for sources of confounding. We formalize conditions under which these latent features enable valid adjustment and statistical inference in ATE estimation, demonstrating results along the example of double machine learning. We discuss critical challenges inherent to latent feature learning and downstream parameter estimation arising from the high dimensionality and non-identifiability of representations. Common structural assumptions for obtaining fast convergence rates with additive or sparse linear models are shown to be unrealistic for latent features. We argue, however, that neural networks are largely insensitive to these issues. In particular, we show that neural networks can achieve fast convergence rates by adapting to intrinsic notions of sparsity and dimension of the learning problem.

## 1. Introduction

Causal inference often involves estimating the average treatment effect (ATE), which represents the causal impact of an exposure on an outcome. Under controlled study setups of randomized controlled trials (RCTs), valid inference methods for ATE estimation are well established (Deaton & Cartwright, 2018). However, RCT data is usually scarce

[1]Department of Statistics, LMU Munich, Munich, Germany [2]Munich Center for Machine Learning (MCML), Munich, Germany. Correspondence to: Rickmer Schulte <schulte@stat.uni-muenchen.de>.

*Proceedings of the 42$^{nd}$ International Conference on Machine Learning*, Vancouver, Canada. PMLR 267, 2025. Copyright 2025 by the author(s).

and in some cases even impossible to obtain, either due to ethical or economic reasons. This often implies relying on observational data, typically subject to (unmeasured) confounding—(hidden) factors that affect both the exposure and the outcome. To overcome this issue of confounding and to obtain unbiased estimates, several inferential methods have been developed to properly adjust the ATE estimation for confounders. One approach that has garnered significant attention in recent years is the debiased/double machine learning (DML) framework (Chernozhukov et al., 2017; 2018), which allows the incorporation of machine learning methods to adjust for non-linear or complex confounding effects in the ATE estimation. DML is usually applied in the context of tabular features and was introduced for ML methods tailored to such features. However, confounding information might only be present in non-tabular data, such as images or text.

**Non-tabular Data as Sources of Confounding** Especially in medical domains, imaging is a key component of the diagnostic process. Frequently, CT scans or X-rays are the basis to infer a diagnosis and a suitable treatment for a patient. However, as the information in such medical images often also affects the outcome of the therapy, the information in the image acts as a confounder. Similarly, treatment and health outcomes are often both related to a patient's files, which are typically in text form. Consequently, ATE estimation based on such observational data will likely be biased if the confounder is not adequately accounted for. Typical examples would be the severity of a disease or fracture. The extent of a fracture impacts the likelihood of surgical or conservative therapy, and the severity of a disease may impact the decision for palliative or chemotherapy. In both cases, the severity will likely also impact the outcome of interest, e.g., the patient's recovery rate. Another famous example is the Simpson's Paradox observed in the kidney stone treatment study of Charig et al. (1986). The size of the stone (information inferred from imaging) impacts both the treatment decision and the outcome, which leads to flawed conclusions about the effectiveness of the treatment if confounding is not accounted for (Julious & Mullee, 1994).

**Contemporary Applications** While the DML framework provides a solution for non-linear confounding, previous

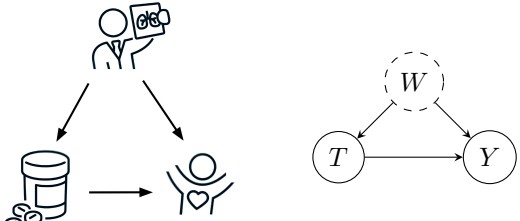

*Figure 1.* Schematic (left) and DAG visualization (right) of the effect of a treatment $T$ on outcome $Y$ that is confounded by non-tabular data $W$ (e.g. information from medical imaging).

examples demonstrate that modern data applications require extending ATE estimation to incorporate non-tabular data. In contrast to traditional statistical methods and classical machine learning approaches, information in non-tabular data usually requires additional feature extraction mechanisms to condense high-dimensional inputs to the relevant information in the data. This is usually done by employing neural network-based approaches such as foundation models or other pre-trained neural networks. While it may seem straightforward to use such feature extractors to extract latent features from non-tabular data and use the resulting information in classical DML approaches, we show that this necessitates special caution. In particular, incorporating such features into ATE estimation requires overcoming previously unaddressed theoretical and practical challenges, including non-identifiability, high dimensionality, and the resulting limitations of standard assumptions like sparsity.

**Problem Setup**  Given $n$ independent and identically distributed (i.i.d.) observations of $(T, W, Y)$, we are interested in estimating the ATE of a binary variable $T \in \{0, 1\}$ on some outcome of interest $Y \in \mathbb{R}$ while adjusting for some source of confounding $W \in \mathbb{W}$ (cf. Figure 1). $W$ is pre-treatment data from some potentially complex sampling space $\mathbb{W}$ that is assumed to be *sufficient* for adjustment. The definition of sufficiency will be formalized in Section 3.1. Under *positivity* and *consistency* assumption—the standard assumptions in causality—the target parameter of interest can be identified as

$$\text{ATE} := \mathbb{E}[\mathbb{E}[Y|T=1, W] - \mathbb{E}[Y|T=0, W]]. \quad (1)$$

While there are many well-known ATE estimators, most require to estimate either the outcome regression function

$$g(t, w) := \mathbb{E}[Y|T=t, W=w] \quad (2)$$

or the propensity score

$$m(t|w) := \mathbb{P}[T=t|W=w] \quad (3)$$

at parametric rate $\sqrt{n}$. Doubly robust estimators such as the Augmented Inverse Probability Weighted, the Targeted

Maximum Likelihood Estimation or the DML approach estimate both *nuisance functions* $g$ and $m$. These methods thus only require the product of their estimation errors to converge at $\sqrt{n}$-rate (Robins & Rotnitzky, 1995; van der Laan & Rubin, 2006; van der Laan & Rose, 2011; Chernozhukov et al., 2017; 2018). However, even this can be hard to achieve, given the *curse of dimensionality* when considering the high-dimensionality of non-tabular data $W$ such as images. Especially given the often limited number of samples available in many medical studies involving images, estimating $m$ and $g$ as a function of $W$, e.g., via neural networks, might not be feasible or overfit easily. To cope with such issues, a common approach is to adopt ideas from transfer learning and use pre-trained neural networks.

**Our Contributions**  In this paper, we discuss under what conditions *pre-trained representations* $Z := \varphi(W)$ obtained from pre-trained neural networks $\varphi$ can replace $W$ in the estimation of nuisance functions (2) and (3). Although the dimensionality of $Z$ is usually drastically reduced compared to $W$, one major obstacle from a theoretical point of view is that representations can only be learned up to invertible linear transformations (e.g., rotations). We argue that common assumptions allowing fast convergence rates, e.g., *sparsity* or *additivity* of the nuisance function, are no longer reasonable in such settings. In contrast, we build on the idea of low *intrinsic dimensionality* of the pre-trained representations. Combining invariance of intrinsic dimensions and functional smoothness with structural sparsity, we establish conditions that allow for sufficiently fast convergence rates of nuisance function estimation and, thus, valid ATE estimation and inference. Our work, therefore, not only advances the theoretical understanding of causal inference in this context but also provides practical insights for integrating modern machine learning tools into ATE estimation.

## 2. Related Work

The DML framework was initially proposed for tabular features in combination with classical machine learning methods (Chernozhukov et al., 2017; 2018). Several theoretical and practical extensions to incorporate neural networks have been made with a focus on tabular data (Shi et al., 2019; Farrell et al., 2021; Chernozhukov et al., 2022; Zhang & Bradic, 2024). Additionally, there is a growing body of research that aims to incorporate non-tabular data as adjustment into DML (Veitch et al., 2019; 2020; Klaassen et al., 2024). While the latter directly incorporates the non-tabular data in the estimation, none of them discuss conditions that would theoretically justify fast convergence rates necessary for valid inference. A different strand of research instead uses either derived predictions (Zhang et al., 2023; Battaglia et al., 2024; Jerzak et al., 2022; 2023a;b) or proxy variables (Kuroki & Pearl, 2014; Kallus et al., 2018; Miao et al., 2018;

Mastouri et al., 2021; Dhawan et al., 2024) in downstream estimation. In contrast to these proposals, we consider the particularly broad setup of using pre-trained representations for confounding adjustment. Given the increasing popularity of pre-trained models, Dai et al. (2022) and Christgau & Hansen (2024) establish theoretical conditions justifying the use of derived representations in downstream tasks, which we will review in the next section. The idea of a low intrinsic dimensionality of non-tabular data and its latent representations to explain the superior performance of deep neural networks in non-tabular data domains has been explored and validated both empirically (Gong et al., 2019; Ansuini et al., 2019; Pope et al., 2021; Konz & Mazurowski, 2024) and theoretically (Chen et al., 2019; Schmidt-Hieber, 2019; Nakada & Imaizumi, 2020). By connecting several of those theoretical ideas and empirical findings, our work establishes a set of novel theoretical results and conditions that allow to obtain valid inference when using pre-trained representations in adjustment for confounding.

## 3. Properties of Pre-Trained Representations

Given the high dimensional nature of non-tabular data, together with the often limited number of samples available (especially in medical domains), training feature extractors such as deep neural networks from scratch is often infeasible. This makes the use of latent features from pre-trained neural networks a popular alternative (Erhan et al., 2010). In order to use pre-trained representations for adjustment in the considered ATE setup, certain conditions regarding the representations are required.

### 3.1. Sufficiency of Pre-Trained Representations

Given any pre-trained model $\varphi$, trained independently of $W$ on another dataset, we denote the learned (last-layer) representations as $Z := \varphi(W)$. Due to the non-identifiability of $Z$ up to certain orthogonal transformations, further discussed in Section 3.2, we define the following conditions for the induced equivalence class of representations $\mathcal{Z}$ following Christgau & Hansen (2024). For this, we abstract the adjustment as conditioning on information in the ATE estimation, namely conditioning on the uniquely identifiable information contained in the sigma-algebra $\sigma(Z)$ generated by any $Z \in \mathcal{Z}$ (see also Appendix A.1 for a special case).

**Definition 3.1.** [Christgau & Hansen (2024)] Given the joint distribution $P$ of $(T, W, Y)$, sigma-algebra $\sigma(Z)$ of $Z$, and $t \in \{0, 1\}$, we say that any $Z \in \mathcal{Z}$ is

(i) $P$-**valid** if:

$$\mathbb{E}_P[\mathbb{E}_P[Y|T = t, \sigma(Z)]] = \mathbb{E}_P[\mathbb{E}_P[Y|T = t, W]]$$

(ii) $P$-**OMS** (Outcome Mean Sufficient) if:

$$\mathbb{E}_P[Y|T = t, \sigma(Z)] = \mathbb{E}_P[Y|T = t, W] \quad (P\text{-a.s.})$$

(iii) $P$-**ODS** (Outcome Distribution Sufficient) if:

$$Y \perp_P W|T, Z.$$

*Remark 3.2.* If $Z \in \mathcal{Z}$ is $P$-ODS, it is also called a *sufficient embedding* in the literature (Dai et al., 2022).

The three conditions in Definition 3.1 place different restrictions on the nuisance functions (2) and (3). While $P$-ODS is most restrictive (followed by $P$-OMS) and thus guarantees valid downstream inference more generally, the strictly weaker condition of $P$-validity is already sufficient (and in fact necessary) to guarantee that $Z \in \mathcal{Z}$ is a *valid adjustment set* in the ATE estimation (Christgau & Hansen, 2024). Thus, any pre-trained representation $Z$ considered in the following is assumed to be at least $P$-valid.

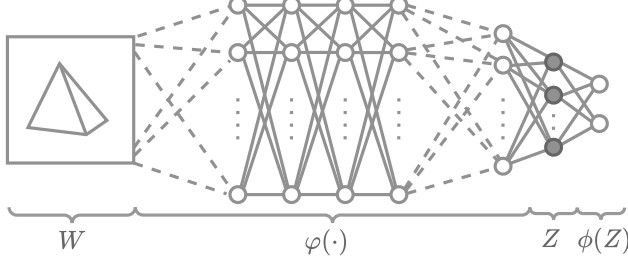

*Figure 2. Schematic visualization of a pre-trained neural network $\varphi(\cdot)$ and representations $Z = \varphi(W)$.*

### 3.2. Non-Identifiability under ILTs

In practice, the representation $Z = \varphi(W)$ is extracted from some layer of a pre-trained neural network $\varphi$. This information does not change under bijective transformations of $Z$, so the representation $Z$ itself is not identifiable. We argue that, in this context, non-identifiability with respect to invertible linear transformations (ILTs) is most important. Suppose $Z = \varphi(W)$ is extracted from a deep network's $\ell$th layer. During pre-training the network further processes $Z$ through a model head $\phi(Z)$, as schematically depicted in Figure 2. The model head usually has the form $\phi^{>\ell}(AZ+b)$ where $A, b$ are the weights and biases of the $\ell$th layer, and $\phi^{>\ell}$ summarizes all following computations. Due to this structure, any bijective linear transformation $Z \mapsto QZ$ can be reversed by the weights $A \mapsto \tilde{A} = AQ^{-1}$ so that the networks $\phi^{>\ell}(A \cdot +b)$ and $\phi^{>\ell}(\tilde{A}Q \cdot +b)$ have the same output.

**Definition 3.3** (Invariance to ILTs). Given a latent representation $Z$, we say that a model (head) $\phi_\xi$ with parameters $\xi \in \Xi$ is non-identifiable up to invertible linear transformations if for any invertible matrix $Q \in \mathbb{R}^{d \times d}$ $\exists \tilde{\xi} \in \Xi : \phi_\xi(QZ) = \phi_{\tilde{\xi}}(Z)$.

Important examples of ILTs are rotations, permutations, and scalings of the feature space as well as compositions thereof.

| Smoothness | + Additivity | + Sparsity & Linearity | Intrinsic Dimension |
|---|---|---|---|
| Stone (1982) | Stone (1985) | Raskutti et al. (2009) | Bickel & Li (2007) |
| $O(n^{-\frac{s}{2s+d}})$ | $O(n^{-\frac{s}{2s+1}})$ | $O(\sqrt{p\log(d)/n}),\, p \ll d$ | $O(n^{-\frac{s}{2s+d_{\mathcal{M}}}}),\, d_{\mathcal{M}} \ll d$ |

*Table 1.* Assumptions and related minimax convergence rates of the estimation error

# 4. Estimation using Pre-Trained Representations

The previous section discussed sufficient and necessary (information theoretic) conditions for pre-trained representations, justifying their usage for adjustment in downstream tasks. The following section will discuss aspects of the functional estimation in such adjustments. Valid statistical inference in downstream tasks usually requires fast convergence of nuisance function estimators. However, obtaining fast convergence rates in high-dimensional estimation problems is particularly difficult. We argue that some commonly made assumptions are unreasonable due to the non-identifiability of representations. We discuss this in the general setting of nonparametric estimation as described in the following.

**The Curse of Dimensionality** The general problem in nonparametric regression is to estimate some function $f$ in the regression model

$$Y = f(X) + \epsilon \tag{4}$$

with outcome $Y \in \mathbb{R}$, features $X \in \mathbb{R}^d$, and error $\epsilon \sim \mathcal{N}(0, \sigma^2)$. The minimax rate for estimating Lipschitz functions is known to be $n^{-\frac{1}{2+d}}$ (Stone, 1982). This rate becomes very slow for increasing $d$, known as the *curse of dimensionality*. Several additional structural and distributional assumptions are commonly encountered to obtain faster convergence rates in high dimensions.

## 4.1. Structural Assumption I: Smoothness

A common structural assumption is the smoothness of the function $f$ in (4), i.e., the existence of $s$ bounded and continuous derivatives. Most convergence rate results assume at least some level of smoothness (see Table 1). The following lemma verifies that this condition is also preserved under ILTs.

**Lemma 4.1** (Smoothness Invariance under ILTs)**.** *Let $D \subseteq \mathbb{R}^d$ be an open set, $f : D \to \mathbb{R}$ be an $s$-smooth-function on $D$, and $Q$ by any ILT. Then $h = f \circ Q^{-1} \colon Q(D) \to \mathbb{R}$ is also $s$-smooth on the transformed domain $Q(D)$.*

The proof of Lemma 4.1 and subsequent lemmas of this section are given in Appendix A.

The lemma shows that a certain level of smoothness of a function defined on latent representations may reasonably be

assumed due to its invariance to ILTs. If the feature dimension is large, however, an unrealistic amount of smoothness would be required to guarantee fast convergence rates (e.g., of order $n^{-1/4}$). This necessitates additional structural or distributional assumptions.

## 4.2. Structural Assumptions II: Additivity & Sparsity

The common structural assumption is that $f$ is *additive*, $f(x) = \sum_{j=1}^{d} f_j(x_j)$, i.e., the sum of univariate $s$-smooth functions. In this case, the minimax convergence rate reduces to $n^{-\frac{s}{2s+1}}$ (Stone, 1985). Another common approach is to rely on the idea of *sparsity*. Assuming that $f$ is $p$-sparse implies that it only depends on $p < \min(n, d)$ features. In case one further assumes the univariate functions to be linear in each feature, i.e. $f(x) = \sum_{j=1}^{p} \beta_j x_j$ with coefficient $\beta_j \in \mathbb{R}$, the optimal convergence rate reduces to $\sqrt{p\log(d/p)/n}$ (Raskutti et al., 2009).

It can easily be shown that the previously discussed conditions are both preserved under permutation and scaling. But as the following lemma shows, sparsity and additivity of $f$ are (almost surely) not preserved under generic ILTs such as rotations.

**Lemma 4.2** (Non-Invariance of Additivity and Sparsity under ILTs)**.** *Let $f : \mathbb{R}^d \to \mathbb{R}$ be a function of $x \in \mathbb{R}^d$. We distinguish between two cases:*

*(i) **Additive**: $f(x) = \sum_{j=1}^{d} f_j(x_j)$, with univariate functions $f_j : \mathbb{R} \to \mathbb{R}$, and at least one $f_j$ being non-linear.*

*(ii) **Sparse Linear**: $f(x) = \sum_{j=1}^{d} \beta_j x_j$, where $\beta_j \in \mathbb{R}$ and at least one (but not all) $\beta_j = 0$.*

*Then, for almost every $Q$ drawn from the Haar measure on the set of ILTs, it holds:*

*(i) If $f$ is additive, then $h = f \circ Q^{-1}$ is not additive.*

*(ii) If $f$ is sparse linear, then $h = f \circ Q^{-1}$ is not sparse.*

Given the non-identifiability of representations with respect to ILTs and the non-invariance result of Lemma 4.2, any additivity or sparsity assumption about the target function $f$ of the latent features seems unjustified. An example of this rotational non-invariance of sparsity is given in Figure 3. This also implies that learners such as the lasso (with

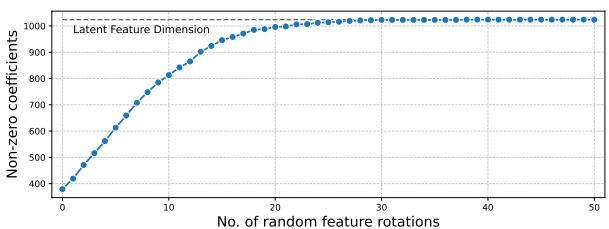

*Figure 3. Non-zero coefficients of a linear classifier on latent features, showing that sparsity is lost with an increasing number of random feature rotations.*

underlying sparsity assumption), tree-based methods that are based on axis-aligned splits (including corresponding boosting methods), and most feature selection algorithms are not ILT-invariant. Further examples can be found in Ng (2004).

### 4.3. Distributional Assumption: Intrinsic Dimension

While the previous conditions are structural assumptions regarding the function $f$ itself, faster convergence rates can also be achieved by making distribution assumptions about the support of $f$. A popular belief is that the $d$-dimensional data $X \in \mathbb{R}^d$ lie on or close to a low-dimensional manifold $\mathcal{M}$ with intrinsic dimension $d_{\mathcal{M}}$. This relates to the famous *manifold hypothesis* that many high-dimensional data concentrate on low-dimensional manifolds (Fefferman et al., 2016, e.g.,). There is strong empirical support for this assumption, especially for non-tabular modalities such as text and images, see Appendix B.1. Given that $d_{\mathcal{M}} \ll d$, and again assuming $f$ to be $s$-smooth, this can lead to a much faster convergence rate of $n^{-\frac{s}{2s+d_{\mathcal{M}}}}$ (Bickel & Li, 2007), as it is independent of the dimension $d$ of the ambient space.

Similarly to Lemma 4.1, the following lemma shows the invariance of the intrinsic dimension of a manifold with respect to any ILT of the coordinates in the $d$-dimensional ambient space.

**Lemma 4.3** (Intrinsic Dimension Invariance under ILTs). *Let $\mathcal{M} \subset \mathbb{R}^d$ be a smooth manifold of dimension $d_{\mathcal{M}} \leq d$. For any ILT $Q$, the transformed set*

$$Q(\mathcal{M}) \, = \, \{\, Qx \mid x \in \mathcal{M} \,\}.$$

*is also a smooth manifold of dimension $d_{\mathcal{M}}$.*

*Remark* 4.4. Put differently, in case the latent representations $Z \in \mathbb{R}^d$ lie on a $d_{\mathcal{M}}$-dimensional smooth manifold $\mathcal{M}$, then the IL-transformed representations $Q(Z)$ also lie on a smooth manifold $Q(\mathcal{M})$ of dimension $d_{\mathcal{M}}$.

Summarizing previous results, the structural and distribution assumptions of smoothness and low intrinsic dimensionality are invariant with respect to any ILT of the features. Hence, as opposed to additivity or sparsity, the two conditions hold not only for a particular instantiation of a latent representa-

tion $Z$ but for the entire equivalence class of latent representations induced by the class of ILTs. This is crucial given the non-identifiability of latent representations, highlighting the importance of low intrinsic dimensions (IDs).

**Deep Networks Can Adapt to Intrinsic Dimensions** Recently, several theoretical works have shown that DNNs can adapt to the low intrinsic dimension of the data and thereby attain the optimal rate of $n^{-\frac{s}{2s+d_{\mathcal{M}}}}$ (Chen et al., 2019; Schmidt-Hieber, 2019; Nakada & Imaizumi, 2020; Kohler et al., 2023). In Section 5, we present a new convergence rate result that builds on the ideas of low ID and a hierarchical composition of functions particularly suited for DNNs.

## 5. Downstream Inference

The manifold assumption alone, however, cannot guarantee sufficient approximation rates in our setting. Even if the manifold dimension $d_{\mathcal{M}}$ is much smaller than the ambient dimension $d$ (for example, $d_{\mathcal{M}} \approx 30$), an unreasonably high degree of smoothness would need to be assumed to allow for convergence rates below $n^{-1/4}$. In what follows, we give a more realistic assumption to achieve such rates. In particular, we combine the low-dimensional manifold structure in the feature space with a structural smoothness and sparsity assumption on the target function.

### 5.1. Structural Sparsity on the Manifold

Kohler & Langer (2021) recently derived convergence rates based on the following assumption.

**Definition 5.1** (Hierarchical composition model, HCM).

*(a) We say that $f \colon \mathbb{R}^d \to \mathbb{R}$ satisfies a HCM of level 0, if $f(x) = x_j$ for some $j \in \{1, \ldots, d\}$.*

*(b) We say that $f$ satisfies a HCM of level $k \geq 1$, if there is a $s$-smooth function $h \colon \mathbb{R}^p \to \mathbb{R}$ such that*

$$f(x) = h\big(h_1(x), \ldots, h_p(x)\big),$$

*where $h_1, \ldots, h_p \colon \mathbb{R}^d \to \mathbb{R}$ are HCMs of level $k-1$.*

*The collection $\mathcal{P}$ of all pairs $(s, p) \in \mathbb{R} \times \mathbb{N}$ appearing in the specification is called the* constraint set *of the HCM.*

An illustration of Definition 5.1 is given in Appendix B.2. The assumption includes the case of sparse linear and (generalized) additive models as a special case but is much more general. Kohler & Langer (2021) and Schmidt-Hieber (2020) exploit such a structure to show that neural networks can approximate the target function at a rate that is only determined by the worst-case pair $(s, p)$ appearing in the constraint set. It already follows from Lemma 4.2 that the

constraint set of such a model is not invariant to ILTs of the input space. Furthermore, the assumption does not exploit the potentially low intrinsic dimensionality of the input space. To overcome these limitations, we propose a new assumption combining the input space's manifold structure with the hierarchical composition model.

**Assumption 5.2.** The target function $f_0$ can be decomposed as $f_0 = f \circ \psi$, where $\mathcal{M}$ is a smooth, compact, $d_\mathcal{M}$-dimensional manifold, $\psi \colon \mathcal{M} \to \mathbb{R}^p$ is $s_\psi$-smooth, and $f$ is a HCM of level $k \in \mathbb{N}$ with constraint set $\mathcal{P}$.

Whitney's embedding theorem (e.g., Lee & Lee, 2012, Chapter 6) allows any smooth manifold to be smoothly embedded into $\mathbb{R}^{2d_\mathcal{M}}$. This corresponds to a mapping $\psi$ with $s_\psi = \infty$ and $p = 2d_\mathcal{M}$ in the assumption above. If not all information in the pre-trained representation $Z$ is relevant, however, $p$ can be much smaller. Importantly, Assumption 5.2 is not affected by ILTs.

**Lemma 5.3** (Invariance of Assumption 5.2 under ILTs). *Let $Q$ be any ILT. If $f_0$ satisfies Assumption 5.2 for a given $\mathcal{P}$ and $(s_\psi, d_\mathcal{M})$, then $\tilde{f}_0 = f_0 \circ Q^{-1}$ satisfies Assumption 5.2 with the same $\mathcal{P}$ and $(s_\psi, d_\mathcal{M})$,*

### 5.2. Convergence Rate of DNNs

We now show that DNNs can efficiently exploit this structure. Let $(Y_i, Z_i)_{i=1}^n$ be i.i.d. observations and $\ell$ be a loss function. Define

$$
f_0 = \underset{f \colon \mathbb{R}^d \to \mathbb{R}}{\arg\min} \, \mathbb{E}[\ell(f(Z), Y)],
$$

$$
\hat{f} = \underset{f \in \mathcal{F}(L_n, \nu_n)}{\arg\min} \, \frac{1}{n} \sum_{i=1}^n \ell(f(Z_i), Y_i),
$$

where $\mathcal{F}(L, \nu)$ is the set of feed-forward neural networks with $L$ layers and $\nu$ neurons per layer. Let $Z \sim P_Z$ and define the $L_2(P_Z)$-norm of a function $f$ as $\|f\|_{L_2(P_Z)}^2 = \int f(z)^2 dP(z)$. We make the following assumption on the loss function $\ell$.

**Assumption 5.4.** There is $a, b \in (0, \infty)$ such that

$$
\frac{\mathbb{E}[\ell(f(Z), Y)] - \mathbb{E}[\ell(f_0(Z), Y)]}{\|f - f_0\|_{L_2(P_Z)}^2} \in [a, b].
$$

Assumption 5.4 is satisfied for the squared and logistic loss, among others (e.g., Farrell et al., 2021, Lemma 8).

**Theorem 5.5.** *Suppose Assumption 5.2 and Assumption 5.4 hold. There are sequences $L_n, \nu_n$ and a corresponding sequence of neural network architectures $\mathcal{F}(L_n, \nu_n)$ such that (up to $\log n$ factors)*

$$
\|\hat{f} - f_0\|_{L_2(P_Z)} = O_p\left( \max_{(s,p) \in \mathcal{P} \cup (s_\psi, d_\mathcal{M})} n^{-\frac{s}{2s+p}} \right).
$$

The result shows that the convergence rate of the neural networks is only determined by the worst-case pair $(s, p)$ appearing in the constraint set of the HCM and the embedding map $\psi$. The theorem extends the results of Kohler & Langer (2021) in two ways. First, it allows for more general loss functions than the square loss. This is important since classification methods are often used to adjust for confounding effects. Second, it explicitly exploits the manifold structure of the input space, which may lead to much sparser HCM specifications and dramatically improved rates.

### 5.3. Validity of DML Inference

In the previous sections, we explored plausible conditions under which the ATE is identifiable, and DNNs can estimate the nuisance functions with fast rates. We now combine our findings to give a general result for the validity of DML from pre-trained representations.

For binary treatment $T \in \{0, 1\}$ and pre-trained representations $Z$, we define the outcome regression function

$$
g(t, z) \coloneqq \mathbb{E}[Y | T = t, Z = z],
$$

and the propensity score

$$
m(z) \coloneqq \mathbb{P}[T = 1 | Z = z].
$$

Suppose we are given an i.i.d. sample $(Y_i, Z_i, T_i)_{i=1}^n$. DML estimators of the ATE are typically based on a cross-fitting procedure. Specifically, let $\bigcup_{k=1}^K I_k = \{1, \ldots, n\}$ be a partition of the sample indices such that $|I_k|/n \to 1/K$. Let $\hat{g}^{(k)}$ and $\hat{m}^{(k)}$ denote estimators of $g$ and $m$ computed only from the samples $(Y_i, Z_i, T_i)_{i \notin I_k}$. Defining

$$
\widehat{\mathrm{ATE}}^{(k)} = \frac{1}{|I_k|} \sum_{i \in I_k} \rho(T_i, Y_i, Z_i; \hat{g}^{(k)}, \hat{m}^{(k)}),
$$

with orthogonalized score

$$
\rho(T_i, Y_i, Z_i; g, m) = g(1, Z_i) - g(0, Z_i)
$$
$$
+ \frac{T_i(Y_i - g(1, Z_i))}{m(Z_i)} + \frac{(1 - T_i)(Y_i - g(0, Z_i))}{1 - m(Z_i)},
$$

the final DML estimate of ATE is given by

$$
\widehat{\mathrm{ATE}} = \frac{1}{K} \sum_{k=1}^K \widehat{\mathrm{ATE}}^{(k)}.
$$

We need the following additional conditions.

**Assumption 5.6.** It holds

$$
\max_{t \in \{0,1\}} \mathbb{E}[|g(t, Z)|^5] < \infty, \quad \mathbb{E}[|Y|^5] < \infty,
$$

$$
\mathbb{E}[|Y - g(T, Z)|^2] > 0, \quad \Pr(m(Z) \in (\varepsilon, 1 - \varepsilon)) = 1,
$$

for some $\varepsilon > 0$.

The first two conditions ensure that the tails of $Y$ and $g(t, Z)$ are not too heavy. The second two conditions are required for the ATE to be identifiable.

**Theorem 5.7.** *Suppose the pre-trained representation is $P$-valid, Assumption 5.6 holds, and the outcome regression and propensity score functions $g$ and $m$ satisfy Assumption 5.2 with constraints $\mathcal{P}_g \cup (s_\psi, d_\mathcal{M})$ and $\mathcal{P}_m \cup (s'_\psi, d_\mathcal{M})$, respectively. Suppose further*

$$\min_{(s,p) \in \mathcal{P}_g \cup (s_\psi, d_\mathcal{M})} \frac{s}{p} \times \min_{(s',p') \in \mathcal{P}_m \cup (s'_\psi, d_\mathcal{M})} \frac{s'}{p'} > \frac{1}{4}, \quad (5)$$

*and the estimators $\hat{g}^{(k)}$ and $\hat{m}^{(k)}$ are DNNs as specified in Theorem 5.5 with the restriction that $\hat{m}^{(k)}$ is clipped away from 0 and 1. Then*

$$\sqrt{n}(\widehat{\text{ATE}} - \text{ATE}) \to \mathcal{N}(0, \sigma^2),$$

*where $\sigma^2 = \mathbb{E}[\rho(T_i, Y_i, Z_i; g, m)^2]$.*

Condition (5) is our primary regularity condition, ensuring sufficiently fast convergence for valid DML inference. It characterizes the necessary trade-off between smoothness and dimensionality of the components in the HCM. In particular, it is satisfied when each component function in the model has input dimension less than twice its smoothness.

# 6. Experiments

In the following, we will complement our theoretical results from the previous section with empirical evidence from several experiments. The experiments include both images and text as non-tabular data, which act as the source of confounding in the ATE setting. Further experiments can be found in Appendix D.

## 6.1. Validity of ATE Inference from Pre-Trained Representations

**Text Data** We utilize the IMDb Movie Reviews dataset from Lhoest et al. (2021) consisting of 50,000 movie reviews labeled for sentiment analysis. The latent features $Z$ as representations of the movie reviews are computed using the last hidden layer of the pre-trained Transformer-based model BERT (Devlin et al., 2019). More specifically, each review results in a 768-dimensional latent variable $Z$ by extracting the [CLS] token that summarizes the entire sequence. For this, each review is tokenized using BERT's subword tokenizer (bert-base-uncased), truncated to a maximum length of 128 tokens, and padded where necessary.

**Image Data** We further use the dataset from Kermany et al. (2018) that contains 5,863 chest X-ray images of children. Each image is labeled according to whether the lung disease pneumonia is present or not. The latent features are

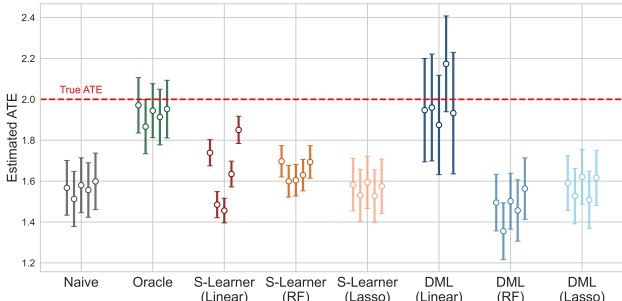

Figure 4. *Label Confounding: Comparison of ATE estimators on the IMDb dataset. DML and S-Learner use pre-trained representations. Point estimates and 95% CIs are depicted.*

obtained by passing the images through a pre-trained convolutional neural network and extracting the 1024-dimensional last hidden layer features of the model. We use the pre-trained Densenet-121 model from the TorchXRayVision library (Cohen et al., 2022), which was trained on several publicly available chest X-ray datasets (Cohen et al., 2020). Further details on the datasets and pre-trained models used in our experiments are provided in Appendix C.1.

**Confounding Setup** For both data applications, we simulate treatment and outcome variables while inducing confounding based on the labels. As an example, for the modified image dataset, children with pneumonia have a higher chance of receiving treatment compared to healthy children. In contrast, pneumonia negatively impacts the outcome variable. The same confounding is present in our modified text dataset. Hence, the label creates a negative bias in both ATE settings if not properly accounted for. Further details about the confounding setups are provided in Appendix C.2.

**ATE Estimators** We compare the performance of DML using three types of nuisance estimators: linear models with and without $L_1$-penalization (Lasso/ Linear), as well as random forest (RF). For comparison, we also include another common causal estimator, called S-Leaner, which only estimates the outcome function (2) (details in Appendix C.3). In each of the simulations, estimators facilitate the information contained in the non-tabular data to adjust for confounding by using the latent features from the pre-trained models in the estimation. As a benchmark, we compare the estimate to the ones of a *Naive* estimator (unadjusted estimation) and the *Oracle* estimator (adjusts for the true label).

**Label Confounding Results** The results for the IMDb experiment over 5 simulations are depicted in Figure 4. As expected, the naive estimator shows a strong negative bias. The same can be observed for the S-Learner (for all nuisance estimators) and for DML using lasso or random forest. In contrast, DML using linear nuisance es-

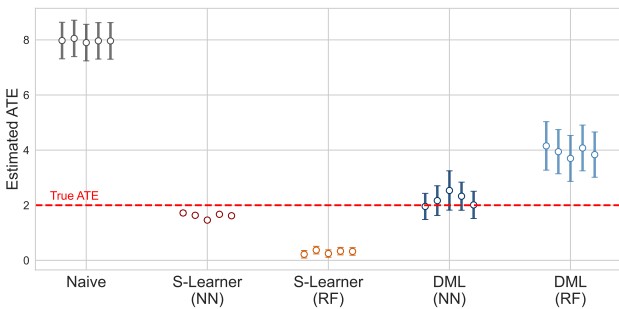

Figure 5. *Complex Confounding: Comparison of ATE estimators on the X-ray dataset. DML and S-Learner use pre-trained representations. Point estimates and 95% CIs are depicted.*

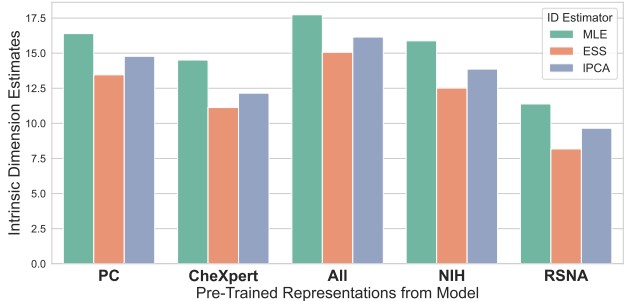

Figure 6. *Different Intrinsic Dimension (ID) estimates of pre-trained representations obtained from different pre-trained models. Representations are based on the X-Ray dataset.*

timators (without sparsity-inducing penalty) yields unbiased estimates with good coverage, as can be seen by the confidence intervals (CIs). First, these results indicate that DML seems to benefit from the doubly robust estimation. Second, DML fails when using ILT non-invariant nuisance estimators such as lasso or random forest. This is because neither of the two can achieve sufficiently fast convergence rates without structural assumptions, such as sparsity or additivity. The latter being unlikely to hold given that representations were shown to be identifiable only up to ILTs. The results for image-based experiment are given in Appendix D.1, where the same phenomenon can be observed.

## 6.2. Neural Networks Adapt to Functions on Low Dimensional Manifolds

In a second line of experiments, we investigate the ability of neural network-based nuisance estimation to adapt to low intrinsic dimensions. The features in our data sets already concentrate on a low-dimensional manifold. For example, Figure 6 shows that the intrinsic dimension of the X-ray images is around $d_{\mathcal{M}} = 12$, whereas the ambient dimension is $d = 1024$. To simulate complex confounding with structural smoothness and sparsity, we first train an autoencoder (AE) with 5-dimensional latent space on the pre-trained representations. These low-dimensional encodings from the AE are then used to simulate confounding. Due to this construction of confounding, the true nuisance functions correspond to encoder-then-linear functions, which are multi-layered hierarchical compositions and therefore align with Assumption 5.2. We refer to this as *complex* confounding.

**Complex Confounding Results** We again compare DML and the S-Learner with different nuisance estimators. In contrast to the previous section, we now use a neural network (with ReLU activation, 100 hidden layers with 50 neurons each) instead of a linear model in the outcome regression nuisance estimation. The results are depicted in Figure 5. Similar to the previous experiments, we find that the naive

estimate is strongly biased similar to the random forest-based estimators. In contrast, the neural network-based estimators exhibit much less bias. While the S-Learner's confidence intervals are too optimistic, the DML estimator shows high coverage and is therefore the only estimator that enables valid inference. The results for the IMDb dataset with complex confounding are given in Appendix D.1.

**Low Intrinsic Dimension** We also investigate the low intrinsic dimension hypothesis about pre-trained representations. Using different intrinsic dimension (ID) estimators such as the Maximum Likelihood (MLE) (Levina & Bickel, 2004), the Expected Simplex Skewness (ESS), and the local Principal Component Analysis (lPCA) we estimate the ID of different pre-trained representations of the X-ray dataset obtained from different pre-trained models from the TorchXRayVision library (Cohen et al., 2022). The results in Figure 6 indicate that the intrinsic dimension of the pre-trained representations is much smaller than the dimension of the ambient space (1024). A finding that is in line with previous research, which is further discussed in Appendix B.1. Additional information on the experiment and the estimators used can be found in Appendix C.4.

## 6.3. The Power of Pre-Training for Estimation

In another line of experiments, we explore the benefits of pre-training in our setup. In particular, we are investigating whether pre-trained neural feature extractors actually outperform non-pre-trained feature extractors in the nuisance estimation of DML-based ATE estimation. We conduct the experiments in the context of the previously introduced image-based *Label Confounding* setup. To adjust for confounding in this setup, nuisance estimators must extract the relevant information from the X-rays. For this purpose, we compare DML using pre-trained feature extractors against DML using neural feature extractors that are trained on downstream data from scratch. While the former uses the same pre-trained Densenet-121 model that was used in previous image-based experiments, the latter incorpo-

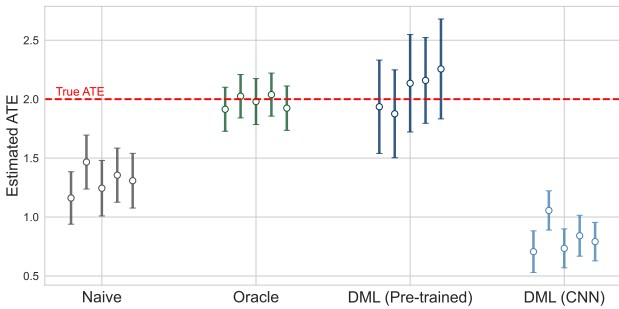

*Figure 7. Comparison of DML using pre-trained representations "DML (Pre-trained)" and DML without pre-training "DML (CNN)" for ATE estimation. Experiment is based on the X-Ray dataset. Point estimates and 95% CIs are depicted.*

rates Convolutional Neural Networks (CNNs) as nuisance estimators into the DML ATE estimation routine. The following experiment is based on 500 sampled images from the X-Ray dataset, where five-layer CNNs are used in the non-pre-trained DML version. Further details about the training and architecture of the utilized CNNs can be found in Appendix C.5.

The results are depicted in Figure 7. For illustrative purposes, we also show the estimates of the *Naive* and *Oracle* estimators, which match those of previous experiments. The key finding of Figure 7 is that DML using pre-trained feature extractors *(DML (Pre-trained))* yields unbiased ATE estimates and well-calibrated confidence intervals, while DML without pre-training *(DML (CNN))* does not. The same phenomenon can be observed in experiments with varying sample sizes and CNN architecture. These experiments are discussed in Appendix D.2. Overall, the results emphasize the benefits of using DML in combination with pre-trained models when utilizing non-tabular data such as images, for confounding adjustment in ATE estimation.

**Further Experiments** Further experiments on the asymptotic normality of DML-based ATE estimation as well as the role of the HCM structure of the nuisance functions are given and discussed in Appendix D.3 and D.4.

## 7. Discussion

In this work, we explore ATE estimation under confounding induced by non-tabular data. We investigate conditions under which pre-trained neural representations can effectively be used to adjust for such kind of confounding. While the representations typically have lower dimensionality, their invariance under orthogonal transformations challenges common assumptions to obtain fast nuisance function convergence rates, like sparsity and additivity. Instead, the study leverages the concept of low intrinsic dimensionality, combining it with invariance properties and structural sparsity to

establish conditions for fast convergence rates in nuisance estimation. This ensures valid ATE estimation and inference, contributing both theoretical insights and practical guidance for integrating machine learning into causal inference.

**Limitations and Future Research** In this work, we focus on a single source of confounding from a non-tabular data modality. A potential future research direction is to study the influence of multiple modalities on ATE estimation. In particular, having multiple modalities requires further causal and structural assumptions on the interplay of the modalities. For example, this could mean that each modality is best processed by a separate network or that the confounding information can only be extracted through a joint network that correctly fuses modalities at some point. We note, however, that this is more of a technical aspect and a matter of domain knowledge, and thus being of minor relevance for the discussion and theoretical contributions of our study.

Moreover, we focused on the estimation of the ATE in this paper, given its popularity in both theory and practice. However, our approach could also be extended to cover other target parameters such as the average treatment effect on the treated (ATT) or the conditional ATE (CATE). While each of these would require a dedicated discussion of the necessary assumptions, we believe that many of the core ideas and results presented here—such as the convergence rates for neural network-based estimation—could also be transferred and used in a theoretical investigation in those settings.

## Impact Statement

This paper presents work whose goal is to advance the field of Machine Learning. There are many potential societal consequences of our work, none which we feel must be specifically highlighted here.

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

# A. Proofs and Additional Results

## A.1. Equivalence Class of Representations

**Lemma A.1** (Equivalence Class of Representations). *Let $(\Omega, \mathcal{F}, P)$ be a probability space, and let $Z : \Omega \to \mathbb{R}^d$ be a measurable map (a random* representation*). Then for each ILT $Q$ the random variable $Q(Z)$ satisfies*

$$\sigma\big(Q(Z)\big) = \sigma(Z),$$

*where $\sigma(Z)$ denotes the $\sigma$-algebra generated by the random variable $Z$. Consequently,*

$$\mathcal{Z} = \big\{Q(Z) \mid Q \in \mathcal{Q}\big\}$$

*forms an equivalence class of representations that are indistinguishable from the viewpoint of measurable information.*

*Proof.* Each $Q \in \mathcal{Q}$ is an invertible linear transformation. Consequently, $Q$ is a Borel measurable bijection with a Borel measurable inverse. To show $\sigma(Q(Z)) = \sigma(Z)$, consider any Borel set $B \subseteq \mathbb{R}^d$. We have

$$\{\omega \in \Omega : Q(Z(\omega)) \in B\} = \{\omega \in \Omega : Z(\omega) \in Q^{-1}(B)\}.$$

Since $Q^{-1}(B)$ is Borel (as $Q$ is a Borel isomorphism), the pre-image $\{\omega : Z(\omega) \in Q^{-1}(B)\}$ belongs to $\sigma(Z)$. Similarly, for any Borel set $A \subseteq \mathbb{R}^d$,

$$\{\omega \in \Omega : Z(\omega) \in A\} = \{\omega \in \Omega : Q(Z(\omega)) \in Q(A)\},$$

which belongs to $\sigma(Q(Z))$. Therefore, $\sigma(Q(Z)) = \sigma(Z)$. $\qquad\square$

## A.2. Proof of Lemma 4.1

*Proof.* We consider $f$ being $C^s$ on the open domain $D \subseteq \mathbb{R}^d$, so by definition, all partial derivatives of $f$ up to order $s$ exist and are continuous on $D$. Further, we consider any invetible matrix $Q$. Such linear transformations are known to be infinitely smooth (as all their partial derivatives of any order exist and are constant, hence continuous). Hence, the function $h = f \circ Q^{-1}$ is the composition of a $C^s$ function $f$ with a linear (and thus $C^\infty$) map $Q^{-1}$.

Applying the multivariate chain rule, we can easily verify that the differentiability properties of $h$ are inherited from those of $f$ and the linear transformation $Q^{-1}$. Specifically, since $Q^{-1}$ is $C^\infty$, and $f$ is $C^s$, their composition $h$ retains the $C^s$ smoothness. Lastly, the (transformed) domain $Q(D)$ is also open as linear (and thus continuous) transformations preserve the openness of sets in $\mathbb{R}^d$. Therefore, $h$ is well-defined and $C^s$ on $Q(D)$. $\qquad\square$

## A.3. Proof of Lemma 4.2

*Proof.* Suppose that $Q$ is an invertible matrix representing the linear map $z \mapsto Q(z)$. Denote by $\tilde{Q} = Q^{-1}$ its inverse and its rows by $\tilde{q}_1, \ldots, \tilde{q}_d$.

**(i) Additivity**

Assume that $f : \mathcal{X} \to \mathbb{R}$ is additive, where $\mathcal{X} \subseteq \mathbb{R}^d$, such that

$$f(x) = \sum_{j=1}^{d} f_j(x_j),$$

and suppose that at least one $f_j$ is nonlinear. Now consider the transformed input space $\tilde{\mathcal{X}} := Q(\mathcal{X}) = \{Qx \mid x \in \mathcal{X}\}$, induced by the invertible linear transformation $Q$. Let $h : \tilde{\mathcal{X}} \to \mathbb{R}$ be given by $h(\tilde{x}) := f(Q^{-1}\tilde{x})$. Then $h$ represents the same mapping as $f$ but expressed in the transformed coordinate system $\tilde{\mathcal{X}}$. In particular, $h(\tilde{x}) = f(x)$, $\forall \tilde{x} \in \tilde{\mathcal{X}}$. Further, we have

$$h(\tilde{x}) = \sum_{j=1}^{d} f_j(\tilde{q}_j^\top \tilde{x}).$$

Assume without loss of generality that $f_1$ is nonlinear. The set of invertible matrices where $\tilde{q}_1$ equals a multiple of a standard basis vector has Haar measure 0. Hence, $f_1(\tilde{q}_1^\top \tilde{x})$ is almost everywhere a nonlinear function of all coordinates of $\tilde{x}$, implying that $h$ is not additive.

**(ii) Sparsity**

Assume $f : \mathcal{X} \to \mathbb{R}$, where $\mathcal{X} \subseteq \mathbb{R}^d$, is sparse linear of the form $f(x) = \beta^\top x$ with $1 \leq \|\beta\|_0 < d$. We again consider the transformed input space $\tilde{\mathcal{X}} := Q(\mathcal{X}) = \{Qx \mid x \in \mathcal{X}\}$, induced by the invertible linear transformation $Q$, and define $h : \tilde{\mathcal{X}} \to \mathbb{R}$ given by $h(\tilde{x}) := f(Q^{-1}\tilde{x})$. Then we have $h(\tilde{x}) = f(Q^{-1}\tilde{x}) = \beta^\top Q^{-1}\tilde{x} =: \tilde{\beta}^\top \tilde{x}$. While the map $h$ is still linear, the set of matrices $Q$ such that $\|\tilde{\beta}\|_0 = \|\beta^\top Q^{-1}\|_0 \neq d$ has Haar measure zero. Hence, $h$ is almost everywhere not sparse. $\qquad\square$

## A.4. Proof of Lemma 4.3

*Proof.* As in the previous proof in Appendix A.2, it is essential to note that ILTs $Q$ are linear, invertible maps that are $C^\infty$ (infinitely differentiable) with inverses that are likewise $C^\infty$. Specifically, $Q$ serves as a global diffeomorphism on $\mathbb{R}^d$, ensuring that both $Q$ and $Q^{-1}$ are smooth ($C^\infty$) functions.

Given that $M$ is a $d_\mathcal{M}$-dimensional smooth manifold, for each point $x$ on the manifold ($x \in M$), there exists a neighborhood $U \subseteq M$ and a smooth chart $\varphi : U \to \mathbb{R}^{d_\mathcal{M}}$ that is a diffeomorphism onto its image. Applying the orthogonal transformation $Q$ to $M$ results in the set $Q(M)$, and correspondingly, the image $Q(U) \subseteq Q(M)$. To construct a smooth chart for $Q(M)$, we can consider the map

$$\tilde{\varphi} : Q(U) \to \mathbb{R}^{d_\mathcal{M}}, \quad \tilde{\varphi}(Q(x)) = \varphi(x),$$

where $x \in U$. Since $Q$ is a diffeomorphism, the composition $\tilde{\varphi} = \varphi \circ Q^{-1}$ restricted to $Q(U)$ remains a smooth diffeomorphism onto its image. Hence, this defines a valid smooth chart for $Q(M)$. Covering $Q(M)$ with such transformed charts derived from those of $M$ ensures that $Q(M)$ inherits a smooth manifold structure. Each chart $\tilde{\varphi}$ smoothly maps an open subset of $Q(M)$ to an open subset of $\mathbb{R}^{d_\mathcal{M}}$, preserving the intrinsic dimension. Therefore, the intrinsic dimension $d_\mathcal{M}$ of the manifold $M$ is preserved under any orthogonal transformation $Q$, and $Q(M)$ remains a $d_\mathcal{M}$-dimensional smooth manifold in $\mathbb{R}^d$. $\qquad\square$

## A.5. Proof of Lemma 5.3

*Proof.* Recall that $Q$ is an invertible linear map, $f_0 = f \circ \psi \colon \mathcal{M} \to \mathbb{R}$, and $\tilde{f}_0 = f_0 \circ \psi \circ Q^{-1} \colon Q(\mathcal{M}) \to \mathbb{R}$. Write $\tilde{f} = f \circ \tilde{\psi}$ with $\tilde{\psi} = \psi \circ Q^{-1} \colon Q(\mathcal{M}) \to \mathbb{R}$. Since $\mathcal{M}$ is a smooth manifold, $Q(\mathcal{M})$ is a smooth manifold with the same intrinsic dimension $d_\mathcal{M}$ by Lemma 4.3. Since $z \mapsto Q^{-1}$ is continuous and $\mathcal{M}$ is compact, $Q(\mathcal{M})$ is also compact. Next, since $\psi$ is $s_\psi$-smooth by assumption, $\tilde{\psi}$ is also $s_\psi$-smooth by Lemma 4.1. Finally, the HCM part $f$ in the two models $f_0$ and $\tilde{f}_0$ is the same, so they share the same constraint set $\mathcal{P}$. This concludes the proof. $\qquad\square$

## A.6. Proof of Theorem 5.5

We will use Theorem 3.4.1 of Van der Vaart & Wellner (2023) to show that the neural network $\hat{f}$ converges at the rate stated in the theorem. For ease of reference, we restate a slightly simplified version of the theorem adapted to the notation used in our paper. Here and in the following, we write $a \lesssim b$ to indicate $a \leq Cb$ for a constant $C \in (0, \infty)$ not depending on $n$.

**Proposition A.2.** *Let $\mathcal{F}_n$ be a sequence of function classes, $\ell$ be some loss function, $f_0$ the estimation target, and*

$$\hat{f} = \arg\min_{f \in \mathcal{F}_n} \frac{1}{n} \sum_{i=1}^n \ell(f(Z_i), Y_i).$$

*Define $\mathcal{F}_{n,\delta} = \{f \in \mathcal{F}_n \colon \|f - f_0\|_{L_2(P_Z)} \leq \delta\}$ and suppose that for every $\delta > 0$, it holds*

$$\inf_{f \in \mathcal{F}_{n,\delta} \setminus \mathcal{F}_{n,\delta/2}} \mathbb{E}[\ell(f(Z), Y)] - \mathbb{E}[\ell(f_0(Z), Y)] \gtrsim \delta^2, \tag{A.2.1}$$

*and, writing $\bar{\ell}_f(z, y) = \ell(f(z), y) - \ell(f_0(z), y)$, that*

$$\mathbb{E}\left[\sup_{f \in \mathcal{F}_{n,\delta}} \left| \frac{1}{n} \sum_{i=1}^n \bar{\ell}_f(Z_i, Y_i) - \mathbb{E}[\bar{\ell}_f(Z, Y)] \right| \right] \lesssim \frac{\phi_n(\delta)}{\sqrt{n}}, \tag{A.2.2}$$

*for functions $\phi_n(\delta)$ such that $\delta \mapsto \phi_n(\delta)/\delta^{2-\varepsilon}$ is decreasing for some $\varepsilon > 0$. If there are $\tilde{f}_0 \in \mathcal{F}_n$ and $\varepsilon_n \geq 0$ such that*

$$\varepsilon_n^2 \gtrsim \mathbb{E}[\ell(\tilde{f}_0(Z), Y)] - \mathbb{E}[\ell(f_0(Z), Y)], \tag{A.2.3}$$

$$\phi_n(\varepsilon_n) \lesssim \sqrt{n}\varepsilon_n^2, \tag{A.2.4}$$

it holds $\|\hat{f} - f_0\|_{L_2(P_Z)} = O_p(\varepsilon_n)$.

*Proof of Theorem 5.5.* Define $(s^*, d^*) = \arg\min_{(s,p) \in \mathcal{P} \cup (s_\psi, d_\mathcal{M})} s/p$ and denote the targeted rate of convergence by

$$\varepsilon_n = \max_{(s,p) \in \mathcal{P} \cup (s_\psi, d_\mathcal{M})} n^{-\frac{s}{2s+p}} (\log n)^4 = n^{-\frac{s^*}{2s^*+d^*}} (\log n)^4.$$

We now check the conditions of Proposition A.2.

**Condition (A.2.1):** Follows from Assumption 5.4, since

$$\inf_{f \in \mathcal{F}_{n,\delta} \setminus \mathcal{F}_{n,\delta/2}} \mathbb{E}[\ell(f(Z), Y)] - \mathbb{E}[\ell(f_0(Z), Y)] \geq \inf_{f \in \mathcal{F}_{n,\delta} \setminus \mathcal{F}_{n,\delta/2}} a\|f - f_0\|_{L_2(P_Z)}^2 \geq \frac{a}{4}\delta^2.$$

**Condition (A.2.2):** Let $N(\varepsilon, \mathcal{F}, L_2(Q))$ be the minimal number of $\varepsilon$-balls required to cover $\mathcal{F}$ in the $L_2(Q)$-norm. Theorem 2.14.2 of Van der Vaart & Wellner (2023) states that eq. (A.2.2) holds with

$$\phi_n(\delta) = J_n(\delta) \left(1 + \frac{J_n(\delta)}{\delta^2\sqrt{n}}\right),$$

where

$$J_n(\delta) = \sup_Q \int_0^\delta \sqrt{1 + \log N(\epsilon, \mathcal{F}(L,\nu), L_2(Q))} d\epsilon,$$

with the supremum taken over all probability measures $Q$. Lemma A.3 in Appendix A.7 gives

$$J_n(\delta) \lesssim \delta\sqrt{\log(1/\delta)} L\nu \sqrt{\log(L\nu)},$$

which implies that $\delta \mapsto \phi_n(\delta)/\delta^{2-1/2}$ is decreasing, so the condition is satisfied.

**Condition (A.2.3):** According to Lemma A.4 in Appendix A.7 there are sequences $L_n = O(\log \varepsilon_n^{-1})$, $\nu_n = O(\varepsilon_n^{-d^*/2s^*})$ such that there is a neural network $\widetilde{f}_0 \in \mathcal{F}(L_n, \nu_n)$ with

$$\sup_{z \in \mathcal{M}} |\widetilde{f}_0(z) - f_0(z)| = O(\varepsilon_n).$$

Together with Assumption 5.4, this implies

$$\mathbb{E}[\ell(\tilde{f}_0(Z), Y)] - \mathbb{E}[\ell(f_0(Z), Y)] \leq b\|\tilde{f}_0 - f_0\|_{L_2(P_Z)}^2 \leq b \sup_{z \in \mathcal{M}} |\widetilde{f}_0(z) - f_0(z)|^2 \lesssim \varepsilon_n^2,$$

as required.

**Condition (A.2.4):** Using $L_n = O(\log \varepsilon_n^{-1})$, $\nu_n = O(\varepsilon_n^{-d^*/2s^*})$ and our bound on $J_n(\delta)$ from Lemma A.3, we get

$$J_n(\delta) \lesssim \delta \log^{1/2}(\delta^{-1}) \varepsilon_n^{-\frac{d^*}{2s^*}} \log^{3/2}(\varepsilon_n^{-1}).$$

Now observe that

$$\begin{aligned}
\frac{\phi_n(\varepsilon_n)}{\varepsilon_n^2} &\lesssim \varepsilon_n^{-\frac{d^*}{s^*}-1} \log^2(\varepsilon_n^{-1}) + \frac{\varepsilon_n^{-\frac{d^*}{s^*}-2} \log^4(\varepsilon_n^{-1})}{\sqrt{n}} \\
&= \varepsilon_n^{-\frac{2s^*+d^*}{2s^*}} \log^2(\varepsilon_n^{-1}) + \varepsilon_n^{-\frac{2s^*+d^*}{s^*}} \log^4(\varepsilon_n^{-1}) n^{-1/2} \\
&\lesssim n^{1/2}(\log n)^{-2} + n^{1/2},
\end{aligned}$$

where the last step follows from our definition of $\varepsilon_n$ and the fact that $\log(\varepsilon_n^{-1}) \lesssim \log n$. In particular, $\varepsilon_n$ satisfies $\phi_n(\varepsilon_n) \lesssim \sqrt{n}\varepsilon_n^2$, which concludes the proof of the theorem. $\square$

### A.7. Auxiliary results

**Lemma A.3.** *Let $\mathcal{F}(L, \nu)$ be a set of neural networks with $\sup_{f \in \mathcal{F}(\mathcal{L}, \nu)} \|f\|_\infty < \infty$. For all $\delta > 0$ sufficiently small, it holds*

$$\sup_Q \int_0^\delta \sqrt{1 + \log N(\epsilon, \mathcal{F}(L, \nu), L_2(Q))} d\epsilon \lesssim \delta \sqrt{\log(1/\delta)} L\nu \sqrt{\log(L\nu)}.$$

*Proof.* Denote by $\mathrm{VC}(\mathcal{F})$ the Vapnik-Chervonenkis dimension of the set $\mathcal{F}$. By Theorem 2.6.7 in Van der Vaart & Wellner (2023), it holds

$$\sup_Q \log N(\varepsilon, \mathcal{F}, L_2(Q)) \lesssim \log(1/\varepsilon) \mathrm{VC}(\mathcal{F}),$$

for $\varepsilon > 0$ sufficiently small. By Theorem 7 of Bartlett et al. (2019), we have

$$\mathrm{VC}(\mathcal{F}(L, \nu)) \lesssim L^2 \nu^2 \log(L\nu).$$

For small $\varepsilon$, this gives

$$\sup_Q \sqrt{1 + \log N(\varepsilon, \mathcal{F}(L, \nu), L_2(Q))} \lesssim \sqrt{\log(1/\varepsilon)} L\nu \sqrt{\log(L\nu)},$$

Integrating the right-hand side gives the desired result. $\qquad\square$

**Lemma A.4.** *Suppose $f_0$ satisfies Assumption 5.2 for a given constraint set $\mathcal{P}$ and $(s_\psi, d_\mathcal{M})$. Define $(s^*, d^*) = \arg\min_{(s,p) \in \mathcal{P} \cup (s_\psi, d_\mathcal{M})} s/p$. Then for any $\varepsilon > 0$ sufficiently small, there is a neural network architecture $\mathcal{F}(L, \nu)$ with $L = O(\log \varepsilon^{-1})$, $\nu = O(\varepsilon^{-d^*/2s^*})$ such that there is $\widetilde{f}_0 \in \mathcal{F}(L, \nu)$ with*

$$\sup_{z \in \mathcal{M}} |\widetilde{f}_0(z) - f_0(z)| = O(\varepsilon).$$

*Proof.* The proof proceeds in three steps. We first approximate the embedding component $\psi$ by a neural network $\widetilde{\psi}$, then the HCM component $f$ by a neural network $\widetilde{f}$. Finally, we concatenate the networks to approximate the composition $f_0 = f \circ \psi$ by $\widetilde{f}_0 = \widetilde{f} \circ \widetilde{\psi}$.

**Approximation of the embedding component.** Recall that $\psi \colon \mathcal{M} \to \mathbb{R}^d$ is a $s_\psi$-smooth mapping. Write $\psi(z) = (\psi_1(z), \ldots, \psi_d(z))$ and note that each $\psi_j \colon \mathcal{M} \to \mathbb{R}$ is also $s_\psi$-smooth. Since $\mathcal{M}$ is a smooth $d_\mathcal{M}$-dimensional manifold, it has Minkowski dimension $d_\mathcal{M}$. Then Theorem 2 of Kohler et al. (2023) (setting $M = \varepsilon^{-1/2s_\psi}$ in their notation) implies that there is a neural network $\widetilde{\psi}_j \in \mathcal{F}(L_\psi, \nu_\psi)$ with $L_\psi = O(\log \varepsilon^{-1})$ and $\nu_\psi = O(\varepsilon^{-d_\mathcal{M}/2s_\psi})$ such that

$$\sup_{z \in \mathcal{M}} |\widetilde{\psi}_j(z) - \psi_j(z)| = O(\varepsilon).$$

Parallelize the networks $\widetilde{\psi}_j$ into a single network $\widetilde{\psi} := (\widetilde{\psi}_1, \ldots, \widetilde{\psi}_d) \colon \mathcal{M} \to \mathbb{R}^d$. By construction, the parallelized network $\widetilde{\psi}$ has $L_\psi$ layers, width $d \times \nu_\psi = O(\nu_\psi)$, and satisfies

$$\sup_{z \in \mathcal{M}} \|\widetilde{\psi}(z) - \psi(z)\| = O(\varepsilon).$$

**Approximation of the HCM component.** Let $a \in (0, \infty)$ be arbitrary. By Theorem 3(a) of Kohler & Langer (2021) (setting $M_{i,j} = \varepsilon^{-1/2p_j^{(i)}}$ in their notation), there is a neural network $\widetilde{f} \in \mathcal{F}(L_f, \nu_f)$ with $L_f = O(\log \varepsilon^{-1})$ and $\nu_f = O(\varepsilon^{-d^*/2s^*})$ such that

$$\sup_{x \in [-a,a]^d} |\widetilde{f}(x) - f(x)| = O(\varepsilon),$$

**Combined approximation.** Now concatenate the networks $\widetilde{\psi}$ and $\widetilde{f}$ to obtain the network $\widetilde{f}_0 = \widetilde{f} \circ \widetilde{\psi} \in \mathcal{F}(L_\psi + L_f, \max\{\nu_\psi, \nu_f\})$. Observe that $L_\psi + L_f = O(\log \varepsilon^{-1})$ and $\nu_\psi + \nu_f = O(\varepsilon^{-d^*/2s^*})$, so the network has the right size. It remains to show that its approximation error is sufficiently small. Define

$$\gamma := \sup_{z \in \mathcal{M}} \|\widetilde{\psi}(z) - \psi(z)\|,$$

which is $O(\varepsilon)$ by the construction of $\widetilde{\psi}$,

$$a := \sup_{z \in \mathcal{M}} \|\psi(z)\| + \gamma,$$

which is $O(1)$ by assumption, and

$$K := \sup_{x, x'} \frac{|f(x) - f(x')|}{\|x - x'\|},$$

which is finite since $f$ is Lipschitz due to $\min_{(s,d) \in \mathcal{P}} s \geq 1$ and the fact that finite compositions of Lipschitz functions are Lipschitz. By the triangle inequality, we have

$$\sup_{z \in \mathcal{M}} |\widetilde{f}_0(z) - f_0(z)| \leq \sup_{z \in \mathcal{M}} |\widetilde{f}(\widetilde{\psi}(z)) - f(\widetilde{\psi}(z))| + \sup_{z \in \mathcal{M}} \|f(\widetilde{\psi}(z)) - f(\psi(z))\|$$

$$\leq \sup_{x \in [-a,a]^d} |\widetilde{f}(x) - f(x)| + K$$

$$= O(\varepsilon),$$

as claimed. $\qquad\square$

## A.8. Proof of Theorem 5.7

*Proof.* We validate the conditions of Theorem II.1 of Chernozhukov et al. (2017). Our Assumption 5.6 covers all their moment and boundedness conditions on $g$ and $m$. By Theorem 5.5, we further know that

$$\|\hat{m}^{(k)} - m\|_{L_2(P_Z)} + \|\hat{g}^{(k)} - g\|_{L_2(P_Z)} = o_p(1).$$

Further, Theorem 5.5 yields

$$\|\hat{m}^{(k)} - m\|_{L_2(P_Z)} \times \|\hat{g}^{(k)} - g\|_{L_2(P_Z)} = O_p\left(\max_{(s,p) \in \mathcal{P}_g \cup (s_\psi, d_\mathcal{M})} n^{-\frac{s}{2s+p}} \times \max_{(s',p') \in \mathcal{P}_m \cup (s'_\psi, d_\mathcal{M})} n^{-\frac{s'}{2s'+p'}}\right)$$

$$= O_p\left(\max_{(s,p) \in \mathcal{P}_g \cup (s_\psi, d_\mathcal{M})} \max_{(s',p') \in \mathcal{P}_m \cup (s'_\psi, d_\mathcal{M})} n^{-\left(\frac{s}{2s+p} + \frac{s'}{2s'+p'}\right)}\right).$$

We have to show that the term on the right is of order $o_p(n^{-1/2})$. Observe that

$$\frac{s}{2s+p} + \frac{s'}{2s'+p'} > \frac{1}{2} \quad \Leftrightarrow \quad \frac{1}{2+p/s} + \frac{1}{2+p'/s'} > \frac{1}{2}$$

$$\Leftrightarrow \quad \frac{4 + p/s + p'/s'}{(2+p/s)(2+p'/s')} > \frac{1}{2}$$

$$\Leftrightarrow \quad 4 + p/s + p'/s' > 2 + p/s + p'/s' + \frac{pp'}{2ss'}$$

$$\Leftrightarrow \quad 4 > \frac{pp'}{ss'}.$$

Thus, our condition

$$\min_{(s,p) \in \mathcal{P}_g \cup (s_\psi, d_\mathcal{M})} \frac{s}{p} \times \min_{(s',p') \in \mathcal{P}_m \cup (s'_\psi, d_\mathcal{M})} \frac{s'}{p'} > \frac{1}{4},$$

implies

$$\|\hat{m}^{(k)} - m\|_{L_2(P_Z)} \times \|\hat{g}^{(k)} - g\|_{L_2(P_Z)} = o_p(n^{-1/2}),$$

as required. $\qquad\square$

# B. Additional Related Literature & Visualizations

## B.1. Empirical Evidence of Low Intrinsic Dimensions

Using different intrinsic dimension (ID) estimators such as the maximum likelihood estimator (MLE; Levina & Bickel, 2004) on popular image datasets such as ImageNet (Deng et al., 2009), several works find clear empirical evidence for low ID of both the image data and related latent features obtained from pre-trained NNs (Gong et al., 2019; Ansuini et al., 2019; Pope et al., 2021). The existence of the phenomenon of low intrinsic dimensions was also verified in the medical imaging (Konz & Mazurowski, 2024) and text-domain (Aghajanyan et al., 2020). All of the mentioned research finds a striking inverse relation between intrinsic dimensions and (state-of-the-art) model performance, which nicely matches the previously introduced theory about ID-related convergence rates.

## B.2. Hierarchical Composition Model (HCM) Visualization

This section provides an illustration of the Hierarchical Composition Model (HCM) that was formally introduced in Definition 5.1. As the name suggests, every HCM is a composition of HCMs of lower level. In Figure 8 we give an illustration of a particular HCM of level 2 and constraint set $\mathcal{P}_{21}$, which we abbreviated by $HCM(2, \mathcal{P}_{21})$. Following the notation of Definition 5.1, the $HCM(2, \mathcal{P}_{21})$ corresponds to the function $f : \mathbb{R}^d \to \mathbb{R}$ defined by $f(x) = h_1^{[2]}(h_1^{[1]}(x), \ldots, h_p^{[1]}(x))$. Each $h_j^{[1]}(x)$ for $j \in \{1, \ldots, p\}$ corresponds to a HCM of level 1, which itself are compositions of HCMs of level 0. Each of the latter corresponds to a feature in the data. The constraint set of each HCM corresponds to the collection of pairs of the degree of smoothness and number of inputs of each HCM it is composed of. For example, assuming that $h_1^{[2]}$ is a $s$-smooth function, then the constraint set of the $HCM(2, \mathcal{P}_{21})$ function $f$ is $\mathcal{P}_{21} = \bigcup_{j=1}^p \mathcal{P}_{1j} \cup (s, p)$. The HCM framework fits both regression and classification. In the latter, the conditional probability would need to satisfy the HCM condition, and any non-linear link function for classification would just correspond to a simple function in the final layer of the HCM.

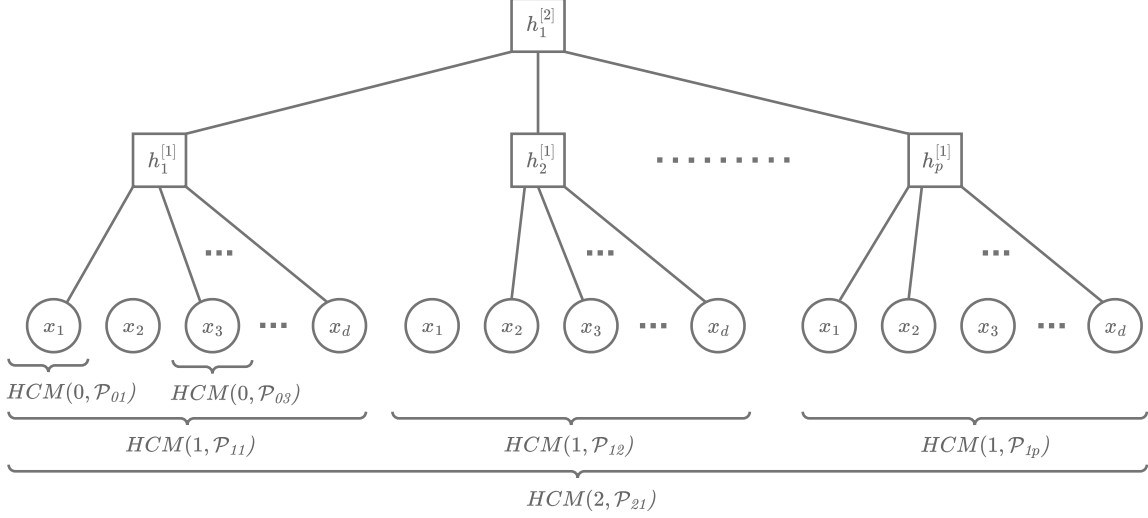

Figure 8. Visualization of a HCM: The illustration depicts a HCM of level 2 and constraint set $\mathcal{P}_{21}$. The HCM of level 2 is a composition of HCMs of level 1, i.e., $h_1, h_2, \ldots, h_p$. The latter are itself compositions of HCMs of level 0, each corresponding to a feature of the data.

# C. Experimental Details and Computing Environment

We conduct several simulation studies to investigate the performance of different Average Treatment Effect (ATE) estimators of a binary treatment on some outcome in the presence of a confounding induced by non-tabular data. In the experiments, the confounding is induced by the labels, i.e., the pneumonia status or the review as well as more complex functions of the pre-trained features. Nuisance function estimation is based on the pre-trained representations that are obtained from passing the non-tabular data through the pre-trained neural models and extracting the last hidden layer features.

## C.1. Data and Pre-trained Models

**IMDb** For the text data, we utilize the IMDb Movie Reviews dataset from Lhoest et al. (2021) consisting of 50,000 movie reviews labeled for sentiment analysis. For each review, we extract the [CLS] token, a 768-dimensional vector per

review entry, of the pre-trained Transformer-based model BERT (Devlin et al., 2019). To process the text, we use BERT's subword tokenizer (bert-base-uncased) and truncate sequences to a maximum length of 128 tokens. We use padding if necessary. After preprocessing and extraction of pre-trained representations, we sub-sampled 1,000 and 4,000 pre-trained representations for the two confounding setups to make the simulation study tractable.

**X-Ray** For the image data simulation, we use the dataset from Kermany et al. (2018) that originally contains 5,863 chest X-ray images of children that were obtained from routine clinical care in the Guangzhou Women and Children's Medical Center, Guangzhou. We preprocess the data such that each patient appears only once in the dataset. This reduces the effective sample size to 3,769 chest X-rays. Each image is labeled according to whether the lung disease pneumonia is present or not. The latent features are obtained by passing the images through a pre-trained convolutional neural network and extracting the 1024-dimensional last hidden layer features of the model. For this purpose, we use a pre-trained Densenet-121 model from the TorchXRayVision library (Cohen et al., 2022). Specifically, we use the model called *densenet121-res224-all*, which is a Densenet-121 model with resolution $224 \times 224$ that was pre-trained on all chest X-ray datasets considered in Cohen et al. (2020). We chose this model for the extraction of pre-trained representation in our experiments, based on its superior performance in benchmark studies conducted in prior work (Cohen et al., 2020). Note that the dataset from the Guangzhou Women and Children's Medical Center that we use, was not used during the training of the model. This is important from a theoretical and practical viewpoint, as the confounding simulation via labels might otherwise be too easy to adjust for given that the model could have memorized the input data. However, using this kind of data we rule out this possibility.

## C.2. Confounding

As introduced in the main text, we simulate confounding both on the true labels of the non-tabular data as well as encodings from a trained autoencoder. While this induces a different degree of complexity for the confounding, the simulated confounding is somewhat similar in both settings. We first discuss the simpler setting of *Label Confounding*. In all of the experiments, the true average treatment effect was chosen to be two.

**Label Confounding** *Label Confounding* was induced by simulating treatment and outcome both dependent on the binary label. In the case of the label being one (so in case of pneumonia or in case of a positive review), the probability of treatment is 0.7 compared to 0.3 when the label is zero. The chosen probabilities guaranteed a sufficient amount of overlap between the two groups. The outcome $Y$ is simulated based on a linear model including a binary treatment indicator multiplied by the true treatment effect (chosen to be 2), as well as a linear term for the label. Gaussian noise is added to obtain the final simulated outcome. The linear term for the label has a negative coefficient in order to induce a negative bias to the average treatment setup compared to a randomized setting. Given that the confounding simulation is only based on the labels, the study was in fact randomized with respect to any other source of confounding.

**Complex Confounding** To simulate *Complex Confounding* with structural smoothness and sparsity, we first train an autoencoder (AE) with 5-dimensional latent space on the pre-trained representations, both in the case of the text and image representations. These AE-encodings are then used to simulate confounding similarly as in the previous experiment. The only difference is that we now sample the coefficients for the 5-dimensional AE-encodings. For the propensity score, these are sampled from a normal distribution, while the sampled coefficients for outcome regression are restricted to be negative, to ensure a sufficiently larger confounding effect, that biases naive estimation. We choose a 5-dimensional latent space to allow for sufficiently good recovery of the original pre-trained representations.

## C.3. ATE Estimators

We estimate the ATE using multiple methods across 5 simulation iterations. In each of these, we estimate a *Naive* estimator that simply regresses the outcome on treatment while not adjusting for confounding. The *Oracle* estimator uses a linear regression of outcome on both treatment and the true label that was used to induce confounding. The S-Learner estimates the outcome regression function $g(t, z) = \mathbb{E}[Y \mid T = t, Z = z]$ by fitting a single model $\hat{g}(t, z)$ to all data, treating the treatment indicator as a feature. The average treatment effect estimate of the S-Learner is then given by

$$\widehat{ATE}_S = \frac{1}{n} \sum_{i=1}^{n} \hat{g}(1, z_i) - \hat{g}(0, z_i).$$

In contrast, the Double Machine Learning (DML) estimators estimates both the outcome regression function and the propensity score to obtain its double robustness property. In our experiments, DML estimators use the partialling-out approach for ATE estimation, which is further discussed in the next paragraph.

In the *Label Confounding* experiments, both the S-Learner and DML estimators are used in combination with linear and random forest-based nuisance estimators. DML (Linear) uses standard linear regression for the estimation of the outcome regression function, and logistic regression with $L_2$-penalty for the estimation of the propensity score. Both nuisance function estimators are ILT-invariant. DML (Lasso) uses $L_1$-penalized linear and logistic regression with cross-validated penalty parameter selection for the outcome regression and propensity score estimation, respectively. The S-Learner (Linear) and S-Learner (Lasso) use unpenalized and $L_1$-penalized linear regression for the outcome regression, respectively. The random forest-based nuisance estimation (both for DML and S-Learner) is based on the standard random forest implementation from *scikit-learn*. The number of estimated trees is varied in certain experiments to improve numerical stability.

In the *Complex Confounding* experiments, we also use neural network-based nuisance estimators for DML and the S-Learner. For this purpose, we employed neural networks with a depth of 100 and a width of 50 while using ReLU activation and Adam for optimization. While DML (NN) and S-Learner (NN) use neural networks for the outcome regression, logistic regression is employed in DML (NN) for propensity score estimation to enhance numerical stability. Generally, DML is used with sample splitting and with two folds for cross-validation. For the S-Learner and DML the Python packages *CausalML* (Chen et al., 2020) and *DoubleML* (Bach et al., 2022) are used, respectively.

**Partially Linear Model and Orthogonal Scores** In our experiments, we simulated two different types of confounding. In both cases we use non-tabular data to adjust for this confounding, given that the confounding inducing information is contained in this data source, but not available otherwise. However, as this information is non-linearly embedded in the non-tabular data, the model that we aim to estimate follows the structure of a so-called *partially linear model* (PLM). Given a binary treatment variable $T$, the PLM is a special case of the more general confounding setup that we consider in the theoretical discussion of this paper. Specifically, the PLM considers the case where the outcome regression function in (2) decomposes as

$$g(T, W) = \mathbb{E}[Y|T, W] = \theta_0 T + \tilde{g}(W). \tag{6}$$

The structure of the propensity score in (3) remains the same. The parameter $\theta_0$ in the PLM corresponds to the target parameter considered in (1), namely the ATE. In their theoretical investigation, Chernozhukov et al. (2018) discuss ATE estimation both in the partially linear model and in the more general setup, which they refer to as *interactive model*. Given that we consider the more general case in Section 5, the orthogonalized score stated in this section matches that of Chernozhukov et al. (2018) for the ATE in the interactive model. In case of the PLM, Chernozhukov et al. (2018) consider two other orthogonalized scores, one of which is the so-called *partialling-out* score function, which dates back to Robinson (1988). The partialling-out score corresponds to an unscaled version of the ATE score in the (binary) interactive model in case the outcome regression decomposes as in (6). The scaling is based on certain estimated weights. Therefore, score functions as the partialling-out score are sometimes referred to as *unweighted* scores (Young & Shah, 2024). While the theoretical result in Theorem 5.7 could also be obtained for DML with partialling-out score under similar assumptions, the key requirement being again (5), the approach may not be asymptotically efficient given that it does not use the efficient influence function. However, the potential loss of asymptotic efficiency is often outweighed by increased robustness in finite-sample estimation when using unweighted scores, which has contributed to the popularity of approaches such as the partialling-out method in practice (van der Vaart (1998, §25.9), Chernozhukov et al. (2018, §2.2.4), Young & Shah (2024)). Accordingly, we also adapted the partialling-out approach in the DML-based ATE estimation in our experiments.

### C.4. Intrinsic Dimensions of Pre-trained Representations

In Section 6.2 we also provide empirical evidence that validates the hypothesis of low intrinsic dimensions of pre-trained representations. For this, we use different pre-trained models from the from the TorchXRayVision library (Cohen et al., 2022). All of these are trained on chest X-rays and use a Densenet-121 (Huang et al., 2017) architecture. Given the same architecture of the models, the dimension of the last layer hidden features is 1024 for all models. The different names of the pre-trained models on the x-axis in Figure 6 indicate the dataset they were trained on. We use the 3,769 chest X-rays from the X-rays dataset described above and pass these through each pre-trained model to extract the last layer features of each model, which we call the pre-trained representations of the data. Subsequently, we use standard intrinsic dimension estimators such as the Maximum Likelihood Estimator (MLE) (Levina & Bickel, 2004), the Expected Simplex Skewness (ESS) estimator (Johnsson et al., 2015), and the local Principal Component Analysis (lPCA) estimator (Fukunaga & Olsen,

1971), with a choice of number of neighbors set to 5, 25 and 50, respectively. While the intrinsic dimension estimates vary by the pre-trained model and the intrinsic dimension estimator used, the results indicated that the intrinsic dimension of the pre-trained representations is much smaller than the dimension of the ambient space (1024).

### C.5. Double Machine Learning with Convolutional Neural Networks as Nuisance Estimators

In Section 6.3, we compare DML with pre-trained neural networks against DML without pre-trained neural networks. This experiment investigates the benefits of pre-training for nuisance estimation in the context of DML-based ATE estimation. Experiments are conducted on the X-Ray dataset, and confounding is simulated based on *Label Confounding*. DML (Pre-trained) uses the same pre-trained Densenet-121 from the TorchXRayVision library (Cohen et al., 2022) that was previously used as pre-trained neural feature extractors in the other image-based experiments. Building on this pre-trained feature extractor, DML (Pre-trained) then uses linear models on the pre-trained features for the nuisance function estimation. In contrast, DML without a pre-trained feature extractor uses standard Convolutional Neural Networks (CNNs) to estimate the nuisance functions directly on top of the images. The experiment of Figure 7 uses a five-layer CNN with $3 \times 3$ convolutions, batch normalization, ReLU activation, and max pooling, followed by a model head consisting of fully connected layers with dropout. Training uses Adam optimization with early stopping. When the network is utilized for propensity score estimation, the outputs are converted to probabilities via a sigmoid activation. Both DML with and without pre-trained feature extractors use the "partialling-out" approach in combination with sample-splitting for doubly-robust ATE estimation.

### C.6. Computational Environment

All computations were performed on a user PC with Intel(R) Core(TM) i7-8665U CPU @ 1.90GHz, 8 cores, and 16 GB RAM. Run times of each experiment do not exceed one hour. The code to reproduce the results of the experiments can be found at `https://github.com/rickmer-schulte/Pretrained-Causal-Adjust`.

## D. Further Experiments

This section provides additional results from experiments that extend those discussed in the main body of the paper.

### D.1. Comparison of ATE Estimators

The results depicted in Figure 9 and Figure 10 complement Figure 4 and Figure 5 that are discussed in Section 6.

**Label Confounding (X-Ray)** The results for the *Label Confounding* simulation based on the X-Ray dataset over 5 simulations are depicted in Figure 9. As before, the naive estimator shows a strong negative bias. Similarly, the S-Learner (for all three types of nuisance estimators) and for DML using random forest or lasso exhibit a negative bias and too narrow confidence intervals. In contrast, DML using linear nuisance estimator (without sparsity-inducing penalty) yields less biased estimates with good coverage due to its properly adapted confidence intervals.

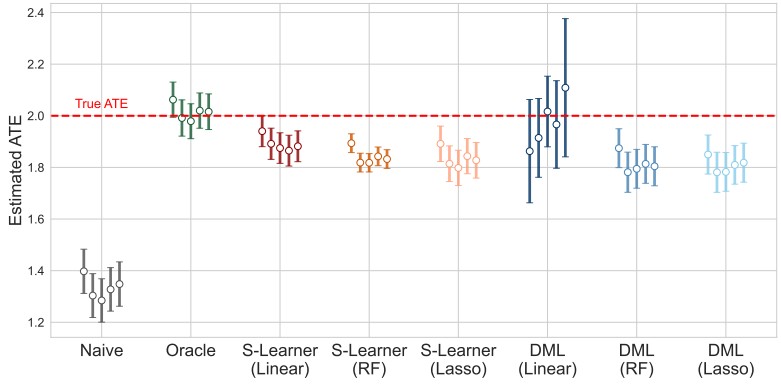

*Figure 9. Label Confounding (X-Ray): Comparison of ATE estimators on the X-Ray dataset. DML & S-Learner use pre-trained representations and three types of nuisance estimators: linear models without $L_1$-penalization (Linear), linear models with $L_1$-penalization (Lasso), as well as random forest (RF). Point estimates and 95% CIs are depicted.*

**Complex Confounding (IMDb)** A similar pattern can be observed for the *Complex Confounding* setting on the IMDb data depicted in Figure 10. The naive estimator and both of the random forest-based ATE estimators exhibit strong bias. In contrast, both neural network-based estimators show very little bias. This provides further evidence that neural networks can adapt to the low intrinsic dimension of the data. However, unlike the DML estimator, the S-Learner still produces overly narrow confidence intervals and thus has poor coverage. As in the example discussed in the main body of the text, the DML (NN) estimator is the only one that yields unbiased estimates and valid inference.

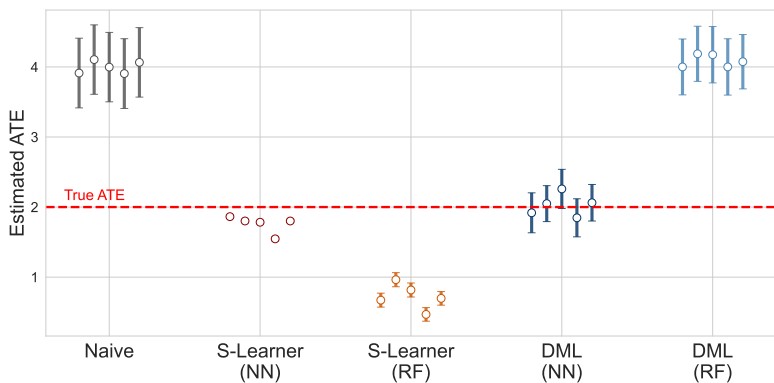

*Figure 10. Complex Confounding (IMDb): Comparison of ATE estimators on the IMDb dataset. DML & S-Learner use pre-trained representations and either neural network (NN) or random forest (RF) based nuisance estimators. Point estimates and 95% CIs are depicted.*

## D.2. DML with and without Pre-Training

The experiment on DML with and without pre-trained representations explored the benefits of pre-training for DML and was discussed in Section 6.3. We extend this line of experiment by considering different sample sizes, as well as neural network architectures for the non-pre-trained model. While the *DML (CNN)* estimator in Figure 7 uses five-layer CNNs for the nuisance function estimation, the *DML (CNN)* estimator in Figure 11 uses a slightly simpler model architecture of two-layer CNNs with ReLU activation and max-pooling, followed by fully connected layers as model head. The slightly less complex model architecture requires fewer neural network parameters to be trained, which might be beneficial in the context of the X-Ray dataset, considering the comparably small sample size available for model training.

Overall, Figure 11 confirms the previous finding that DML with pre-trained representations performs much better than DML without pre-training. While the former yields unbiased ATE estimates, the ATE estimates of the latter show a strong negative bias. As the two plots show, this result is independent of the model architecture and sample size used.

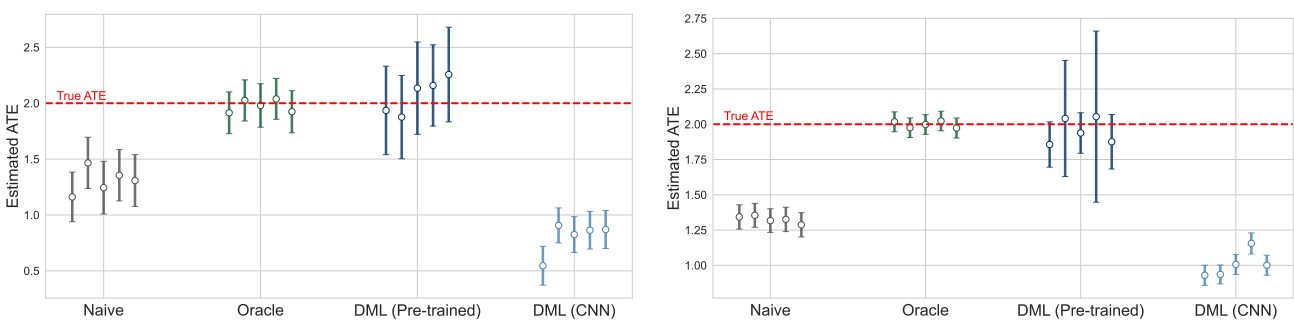

*Figure 11. DML with/out Pre-Training: Comparison of DML using pre-trained representations "DML (Pre-trained)" and DML without pre-training "DML (CNN)" for ATE estimation using 500 (Left) and all 3769 (Right) images from the X-Ray dataset. Point estimates and 95% CIs are depicted.*

### D.3. Asymptotic Normality of DML ATE Estimation

This line of experiments explores the asymptotic normality of the DML estimator. For this purpose, we extend the ATE estimation experiments of Figure 9 with *Label Confounding* and Figure 5 with *Complex Confounding* that are based on the X-Ray dataset. While the two figures depict the estimates of different ATEs over 5 simulation iterations, we repeat both experiments with 200 iterations and collect the ATE estimates. We standardize each estimate and plot the corresponding empirical distribution. The results are shown in Figure 12. The left plot depicts the empirical distribution of the 200 standardized point estimates of the *Oracle* and *Naive* estimator, as well as *DML* with linear nuisance estimator in the *Label Confounding* experiment. The right plot displays the empirical distribution of the 200 standardized point estimates of the *Naive*, *S-Learner*, and *DML* estimator in the *Label Confounding* experiment. The latter two use neural network-based nuisance estimation. While the distributions of the *Naive* and *S-Learner* estimators show a strong bias, the distribution of the DML approach matches the theoretical standard normal distribution in both experiments.

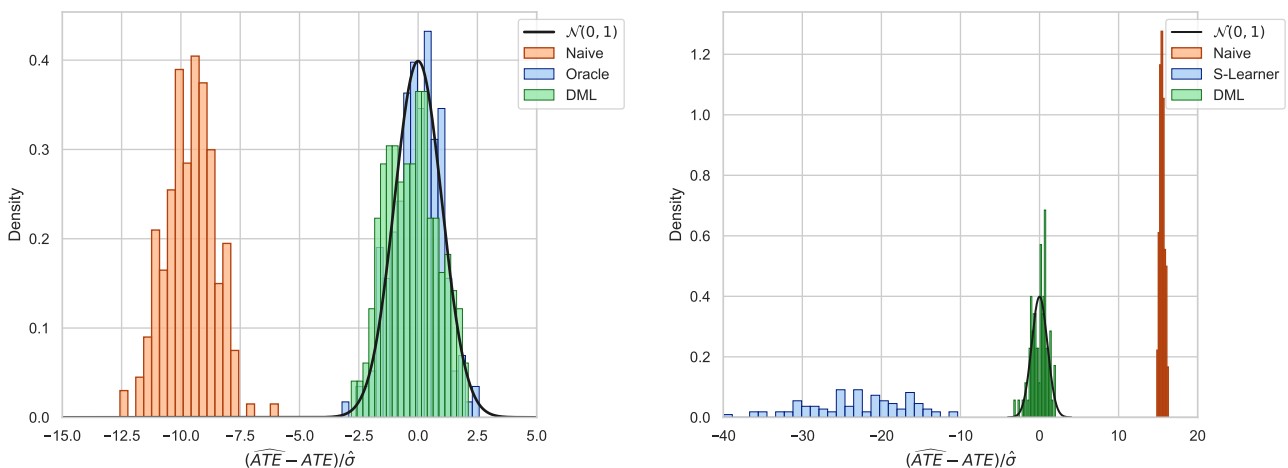

Figure 12. *Asymptotic Normality of DML: Comparison of the empirical distributions of standardized point estimates of different ATE estimators on the X-ray dataset. Left: Distribution of 200 standardized point estimates of the Naive, Oracle, and DML with linear nuisance estimation from the Label Confounding experiment. Right: Distribution of 200 standardized point estimates of the Naive, S-Learner with NN-based nuisance estimation (S-Learner) and DML with NN-based nuisance estimation from the Complex Confounding experiment.*

### D.4. Effects of the Hierarchical Composition Model (HCM) Structure on Estimation

In this experiment, we investigate the effect of the Hierarchical Composition Model (HCM) structure in the context of ATE estimation. The HCM was formally introduced in Definition 5.1 and later used in Theorem 5.5 to derive convergence rates of neural network-based estimation. This result showed that the convergence rate of neural networks is determined by the worst-case pair that appeared in the constraint set of the HCM. The core benefit of the HCM in our context is, that it constitutes a very flexible class of functions, while at the same time, it enables to obtain fast convergence rates in case the target function factors favorably according to the hierarchical structure of the HCM. The latter could be fulfilled in case each composition in the hierarchical structure only depends on a few prior compositions. This structural sparsity would improve the worst-case pair in the constraint set and thereby could allow for obtaining fast convergence rates.

Now, in the DML ATE estimation, one might be interested in what happens when the nuisance functions do not factor according to the HCM structure, such that sufficiently fast convergence rates for the nuisance estimation, required in the ATE estimation, can be achieved. We simulate such a scenario by extending the previously introduced setup of *Complex Confounding*. Instead of simulating confounding based on low-dimensional encodings from a AE trained on the pre-trained representations from the X-Ray experiments (as done in some of the previous experiments), we do this directly based on the pre-trained representations. For this purpose, we define the nuisance functions (outcome regression and propensity score) to depend on the *product of all features* in each pre-trained representation. Hence, the nuisance functions depend on all 1024 features, thereby mimicking the *curse of dimensionality* scenario discussed in Section 4. Further, the nuisance functions are constructed such that bias is introduced in the ATE estimation. In the following, we estimate the same set of ATE estimators that were previously used in the *Complex Confounding* experiments.

The results are depicted in Figure 13. All estimators show substantial bias (even the DML approach), given that none of the estimators is able to properly adapt to this complex type of confounding structure. The results are a validation of the fact that no (nuisance) estimator can escape the *curse of dimensionality* without utilizing certain beneficial structural assumptions.

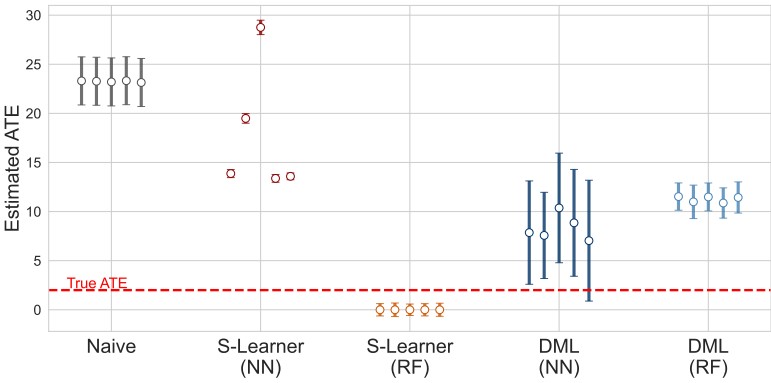

Figure 13. Comparison of ATE estimators: DML & S-Learner use pre-trained representations and either neural network (NN) or random forest (RF) based nuisance estimators. Point estimates and 95% CIs are depicted.

