# OpenReview forum: "Adjustment for Confounding using Pre-Trained Representations"
_ICML.cc/2025/Conference — ICML 2025 poster_

### Official Review · Reviewer_UvN8 · 2025-03-11

**Overall Recommendation:** 3

**Summary:**

This paper explores how non-tabular data, such as images and text, can be incorporated into average treatment effect (ATE) estimation to account for confounding factors. The authors propose using latent features from pre-trained neural networks for adjustment. They formalize conditions under which these features enable valid ATE estimation, particularly in the double machine learning framework. The paper highlights challenges related to high-dimensional representations and non-identifiability but argues that neural networks can overcome these issues by adapting to intrinsic sparsity and dimensional structures, enabling fast convergence rates in treatment effect estimation.

**Claims And Evidence:**

I found this paper to be a primarily theoretical paper with minimal experimental support. For example, the last few arguments in the abstract are "Common structural assumptions for obtaining fast convergence rates with additive or sparse linear models are shown to be unrealistic for latent features. We argue, however, that neural networks are largely insensitive to these issues. In particular, we show that neural networks can achieve fast convergence rates by adapting to intrinsic notions of sparsity and dimension of the learning problem." but I couldn't find experimental  support for these claims.

Moreover, just like the paper, I find the formulation of the problem unpractical. Indeed, imaging data could be regarded as confounders but practically modeling this does not really gain you any improved understanding. All examples given in the paper have a simplistic form; i.e., the severity of disease, the size of the stone, or the extent of a fracture is a univariate confounder. I'm all for modeling these clear and interpretable confounding effects, but using images as their surrogate seems to be a detour and over-complication. In the end, we don't know whether image latent representations (extracted by a pre-trained network) actually contain those information or not.

**Essential References Not Discussed:**

NA

**Experimental Designs Or Analyses:**

* I feel the experimental setups are over simplified. Having a simple 0.7/0.3 ratio for simulating the basic label confounding and the 5-dimensional autoencoder latent representation for complex confounding have very limited generalizability to real life problems given the confounding effects in imaging applications are high-dimensional in nature.

* Compared to other estimators, the proposed one does not seem to be better in Fig. 4&5.

**Methods And Evaluation Criteria:**

Methods:

The formulation and theoretical derivation look correct to my eye, but I have to admit that I'm not a statistician so might not be able to fully appreciate the methodological and theoretical sophistication. I made my overall assessment assuming great merit in these formulations.

Evaluation:

The shortcoming of the study is in the evaluation. The core analysis is two plots (Fig. 4&5) showing the estimated ATE by the proposed estimator and some baseline estimators. These two experiments are toy examples in nature and have extremely simple setup. Even confined to this experimental settings, the study could explore way more different simulation scenarios, e.g., by varying simulation parameters.

**Other Comments Or Suggestions:**

NA

**Other Strengths And Weaknesses:**

NA

**Questions For Authors:**

NA

**Relation To Broader Scientific Literature:**

Yes, the paper presents the scenario that they want to dive into, which is a specific case in the broader confounder analysis literature.

**Theoretical Claims:**

As mentioned above, The formulation and theoretical derivation look correct to my eye but I don't have the expertise to fully assess its correctness.

---

> ### Author Rebuttal · Authors · 2025-04-01
>
> **Additional experiments** can be found at https://anonymous.4open.science/r/icml2025-6599/add_exp_r3.pdf
>
> - - - -
>
> ### **Theory & Practical Relevance**
>
> > I found this paper to be a primarily theoretical paper with minimal experimental support.
>
> This is correct. Our paper is a theoretical contribution to the fast-growing literature that aims to incorporate non-tabular data and pre-trained models in causal inference procedures. While many papers do this empirically (as described in Sec. 2), we establish a novel set of theoretical conditions allowing for valid statistical inference of ATE estimation in this context. Our experiments mainly serve to illustrate these concepts.
>
> Nonetheless, we followed the reviewer’s suggestion and added several additional experiments (see answer below).
>
> > [NNs] can achieve fast convergence rates by adapting to intrinsic notions of sparsity and dimension of the learning problem." [...] I couldn't find experimental support for these claims.
>
> We provide empirical support for these claims in our experiments on complex confounding shown in Fig. 5 & 8. The confounding via latent features precisely mimics the idea of low intrinsic dimension and HCM structure of the target function. In contrast to the other depicted ATE estimators, NN-based nuisance estimation denoted by “DML (NN)” shows that NNs can adapt to the intrinsic notions of sparsity and dimension of the learning problem, thereby yielding unbiased ATE estimation. We thank the reviewer for the remark and will emphasize this in the revised version of the paper.
>
> > I find the formulation of the problem unpractical.[...] practically modeling [imaging data] does not really gain you any improved understanding [...] images as their surrogate seems to be a detour and over-complication.
>
> We politely disagree with this viewpoint. Incorporating non-tabular data in ATE estimation to adjust for confounding does strongly impact scientific understanding and resembles what is done in practical application, as also described in our motivating examples in Sec. 1. In scenarios, where the confounder is available only embedded in non-tabular data (e.g. kidney stone or tumor size can only be measured via medical imaging), incorporating such modality in the ATE estimation is crucial to obtain unbiased estimates and draw valid scientific conclusions, as we show both theoretically and empirically in our paper.
>
> - - - -
>
> ### **Empirical Evaluation & Additional Experiments**
>
> > Experimental setups are over simplified. Having [...] a [5-dim. AE latent rep.] for complex confounding have very limited generalizability to real life problems given the confounding effects in imaging applications are high-dimensional in nature.
>
> We thank the reviewer for raising this point. However, all of the previously mentioned confounding “in nature”, e.g. kidney stone or tumor size, is in fact low dimensional yet embedded in the medical image in high dimensions, making it necessary to extract this information. Our experiments using both text and X-ray data precisely show that pre-trained models can be used to extract the latent confounding (low-dim.) information from the high-dim. non-tabular data modalities and achieve valid inference.
>
> > The study could explore way more different simulation scenarios, e.g., by varying simulation parameters.
>
> Based on the reviewer's suggestion, we conducted several additional experiments, in which we varied simulation parameters including the treatment assignment probabilities and the size of the latent dim. of the confounder.  The new results (Fig. 9-12) are in line with our previous results and reinforce our main theoretical findings. In particular, results are insensitive to changes in assignment prob. and latent dim. We thank the reviewer for this remark and think these results further strengthen our paper. Following the suggestion of reviewer vZqT we also conducted several other experiments, e.g. comparing DML with and without pre-training in Fig. 14.
>
> > Compared to other estimators, the proposed one does not seem to be better in Fig. 4&5.
>
> This might be a misunderstanding. The estimators suggested by us (DML Linear and the DML NN) are the only estimators in Figs. 4 & 5 that yield unbiased ATE estimation with good coverage (overlap of confidence intervals with the true ATE, i.e., the red line) while the other estimators do not. Note that the unbiased “Oracle” estimator in Fig. 4 is an infeasible model in practice and serves as a gold standard comparison.
>
> - - - -
>
> We thank the reviewer for the suggestions related to additional experiments. We hope the additional experiments in our response address the points raised in the review. Should the reviewer find the response satisfactory, we would appreciate reconsidering the initial score. Otherwise, we remain fully committed to addressing any remaining concerns during the second author response phase.

---

> > ### Comment · Reviewer_UvN8 · 2025-04-02
> >
> > Thanks for the rebuttal. The explanation helps my understanding. I'm on the fence. I still cannot wrap my head around why we need to incorporate images in ATE estimation. I'm all for correcting for kidney stone size or tumor size, but why can't we just have a model estimating those measures and use those tabular measures instead? Put in other words, are we sure that those latent variables contain the confounding information of interest? I know this is a theoretical piece, but I just want to make sure we are solving a real problem.

---

> > > ### Author Response · Authors · 2025-04-03
> > >
> > > We thank the reviewer for allowing us to elaborate on these aspects in further detail.
> > >
> > > > Why [do] we need to incorporate images in ATE estimation. I'm all for correcting for kidney stone size or tumor size, but why can't we just have a model estimating those measures and use those tabular measures instead?
> > >
> > > - In fact, what the reviewer is suggesting can be regarded as a special case of what we do. We use a model to estimate a tabular representation of the confounding information in the image. As a special case, this could also correspond to predicting confounders directly.
> > > - If all relevant confounders were known, but the information in the data is only contained in the image, it would indeed be possible to design specific models to account for them, as the reviewer correctly points out. For this, we suggest a pre-trained model.
> > > -  In observational studies, relevant confounders are not always known a priori to the analyst. In this case, the additional advantage of using pre-trained representations is that they can account for many other potential confounders beyond pre-selected or manually predicted confounders (the Densenet-121 model used in our application, for example, was trained to detect 18 different anomalies on the chest X-ray scans).
> > >
> > > > Put in other words, are we sure that those latent variables contain the confounding information of interest?
> > >
> > > - Of course, we cannot be sure in general. The situation is no different from other ATE estimation situations where the  “no unmeasured confounding” assumption is unavoidable. This is more precisely characterized in Def 3.1 (i) in our context.
> > > - However, this assumption becomes more reasonable in our context, given that pre-trained representations encompass a variety of potential confounding information about the image and potentially much more than what manually predicted confounders would contain (as mentioned above).
> > > - On the other hand, if the tabular data has no record of the relevant confounding information (e.g., the tumor size), the only option is to estimate it from the non-tabular (image) data source.
> > >
> > > > I know this is a theoretical piece, but I just want to make sure we are solving a real problem.
> > >
> > > - We understand this concern, but we would like to point out that we are by no means the first to investigate the problem of incorporating non-tabular data in ATE estimation. This approach has been explored by different studies in many real-world examples (e.g. Veitch et al., 2019, 2020; Jerzak et al., 2022 a,b, 2023; Klaassen et al., 2024; Dhawan et al., 2024). Our core contribution is embedding these practical approaches in a broader theoretical framework, highlighting potential pitfalls, and providing theoretical guarantees for valid statistical inference.
> > > - In the upcoming years, several new open EHR databases (e.g., MIMIC-IV or European Health Data Space) will make a plethora of data (including non-tabular data) publicly available, making scenarios of estimating ATE from real-world observational data settings while requiring adjustment for non-tabular data even more relevant.
> > >
> > > We once again thank the reviewer for taking the time to engage with our responses and hope we have addressed all remaining concerns.
> > >
> > > ----
> > >
> > > **References**
> > > - Dhawan et al. (2024), End-to-end causal effect estimation from unstructured natural language data
> > > - Klaassen et al. (2024), DoubleMLDeep: Estimation of causal effects with multimodal data
> > > - Jerzak et al. (2023), Integrating earth observation data into causal inference: challenges and opportunities
> > > - Jerzak et al. (2022), Estimating causal effects under image confounding bias with an application to poverty in africa
> > > - Jerzak et al. (2022), Image-based treatment effect heterogeneity
> > > - Veitch et al. (2020), Adapting text embeddings for causal inference
> > > - Veitch et al. (2019), Using embeddings to correct for unobserved confounding in networks

---

### Official Review · Reviewer_gvFH · 2025-03-13

**Overall Recommendation:** 3

**Summary:**

The paper revisits the problem of estimating Average Treatment Effects (ATE) in observational studies under an assumption of ignorability where confounding factors are available as images or text (non-tabular data). The authors develop a theoretical argument that describes under what conditions the use of pre-trained neural networks to extract relevant features (representations) from non-tabular data gives valid adjustment within the Double Machine Learning (DML) framework. One key theoretical contribution is to show that the intrinsic dimension of the latent representation Z is invariant under linear transformations which combined with properties of the target function f(z) leads to fast convergence of the ATE.

**Claims And Evidence:**

All claims are very well supported both theoretically and empirically.

**Essential References Not Discussed:**

No significant reference appears to be missing.

**Experimental Designs Or Analyses:**

Yes, experimental design is sound.

**Methods And Evaluation Criteria:**

Yes, all good.

**Other Comments Or Suggestions:**

See above.

**Other Strengths And Weaknesses:**

The paper is very well executed: it introduces existing convergence guarantees for function approximation, highlights the challenges well, and provides a compelling argument for the validity for using pre-trained features for adjustment.

The HCM definition is a little bit difficult to visualize. Could more details or examples be given to better convey the intuition.

It seems to me that a lot of the heavy lifting is done in the function approximation theorems of Secs. 4 and 5.1 and 5.2. Once those are established ATE estimation follows more or less straightforwardly. Why frame the results so closely to ATE estimation? Is there not an opportunity here to provide guarantees for any functional of pre-trained representations?

With my causality hat on, I would say that the biggest challenge for the correctness of the procedure is to guarantee Def. 3.1. It is highly non-trivial that the representation will accurately preserve the information content in the original data. (I guess this is a limitation of all DML work so might not need addressing here.)

**Questions For Authors:**

Sec. 5.2, in which f is parameterized by a feed forward NN, is not illustrated in the experiments (as far as I know). The authors instead use linear functions and random forest regressors. Is there a reason for this disconnect between theory and experiments? If no non-linearities can be incorporated into the target function parameterization then that is a potential limitation of the theory.

Classification problems typically require a non-linear transformation or a link function, do classification functions satisfy the HCM condition?

**Relation To Broader Scientific Literature:**

Yes, related work is discussed appropriately.

**Theoretical Claims:**

Did not check proofs in detail but follows established results in the literature.

---

> ### Author Rebuttal · Authors · 2025-04-01
>
> ### **Theoretical Results**
>
> > Why frame the results so closely to ATE estimation? Is there not an opportunity here to provide guarantees for any functional of pre-trained representations?
>
> Indeed, the derived convergence rates could be used for convergence guarantees of any functional of the pre-trained representations and thereby establish asymptotic results about other causal estimands other than the ATE. Given the popularity of the ATE (both in theory and practice), we chose to focus on the ATE estimand in the context of DML. However, as correctly pointed out by the reviewer, our convergence rates can be used more generally, e.g., to provide asymptotic inference results for estimands such as the average treatment effect of the treated (ATT) and the conditional ATE (CATE). For the latter, however, stronger assumptions would be required, e.g., $P$-OMS instead of $P$-valid pre-trained representations.
>
> Based on the reviewer’s excellent remark, we will add a discussion to Sec. 5 and elaborate on this aspect.
>
> > I would say that the biggest challenge for the correctness of the procedure is to guarantee Def. 3.1. It is highly non-trivial that the representation will accurately preserve the information content in the original data. (I guess this is a limitation of all DML [...])
>
> We thank the reviewer for raising this relevant point. Indeed, validating the assumption of Def. 3.1 in practice is not trivial — as with any other causal assumption. However, this limitation is not specific to DML per se, but is instead inherent to any method aiming to estimate the ATE based on pre-trained representation for adjustment, given that Def. 3.1 (i) is necessary for the identification of the ATE in this context. We will make this clear in a revised version of the paper and thank the reviewer for bringing this up.
>
> > Sec. 5.2, in which f is parameterized by a feed forward NN, is not illustrated in the experiments (as far as I know). The authors instead use linear functions and random forest regressors. Is there a reason for this disconnect between theory and experiments?
>
> This seems to be a misunderstanding. In all experiments, we use DML with neural networks as regressors. The “Label confounding” setup in Figs. 4 & 7 is simple enough that a single layer (“linear”) suffices; in the “Complex confounding” setup in Figs. 5 & 8, we use a 100-layer neural network. The RF regressor is included to highlight that methods not invariant to ILT transformations often fail when used on pre-trained representations.
>
> Hence, our experiments are well in line with our theory. We will state this more clearly in the revised version.
>
> ----
>
> ### **HCM**
>
> > Classification problems typically require a non-linear transformation or a link function, do classification functions satisfy the HCM condition?
>
> Yes, this is correct. More precisely, in the classification context, the conditional probability would satisfy the HCM condition. A non-linearity or link is not a problem in this context, since it is just a simple function in the final layer of the HCM. We thank the reviewer for this question and will clarify this in the revised version of the paper.
>
> > The HCM definition is a little bit difficult to visualize. Could more details or examples be given to better convey the intuition.
>
> We thank the reviewer for this remark. To better illustrate the concept of the HCM in our paper, we will add a visualization of the HCM similar to the one from the original publication (Kohler and Langer, Annals of Statistics, 2021) and add further explanation to it.
>
> - - - -
>
> We appreciate the reviewer's thoughtful remarks and important points related to the assumptions in our paper. We hope the additional clarifications in our response address the points raised in the review. Should the reviewer find the response satisfactory, we would appreciate reconsidering the initial score. Otherwise, we remain fully committed to addressing any remaining concerns during the second author response phase.

---

### Official Review · Reviewer_vZqT · 2025-03-14

**Overall Recommendation:** 3

**Summary:**

This paper investigates the application of Double Machine Learning (DML) for estimating the Average Treatment Effect (ATE) in non-tabular data contexts, such as text and images. The authors highlight the limitations of traditional causal inference methods in handling non-tabular data and propose leveraging pre-trained representations from deep neural networks (DNNs) to adjust for confounding variables.

**Claims And Evidence:**

Yes

**Essential References Not Discussed:**

No, the related works are thoroughly summarized and appropriately referenced.

**Experimental Designs Or Analyses:**

The experimental setup is well-structured, with a clear dataset and a relevant benchmark.

**Methods And Evaluation Criteria:**

Yes

**Other Comments Or Suggestions:**

-Consider adding empirical tests on ILTs to evaluate their impact on DML estimation.

-Clarify how HCM enhances DML performance beyond theoretical claims, providing empirical evidence if possible.

**Other Strengths And Weaknesses:**

**Strengths**:

-Novel application of DML to non-tabular data.

-Theoretical contributions on ILTs and HCM are interesting and relevant.

**Weaknesses**:

-The manuscript lacks empirical validation of the impact of ILTs on ATE estimation.

-While the benefits of HCM are theoretically motivated, they are not directly tested or empirically verified.

-Critical ablation studies, such as comparing DML with raw data versus pre-trained features, are missing.

-Although ILTs are extensively discussed, their connection to the experimental results is not clearly established or well-articulated.

**Questions For Authors:**

1. How does ILT invariance specifically impact DML estimation?

2. Why is there no experiment comparing DML with and without pre-trained representations?

3. Are there scenarios where HCM does not hold?bIf real-world data does not follow an HCM structure, how does this affect ATE estimation?

**Relation To Broader Scientific Literature:**

The work relates to prior research on DML for causal inference, particularly in tabular data settings (Chernozhukov et al., 2017; 2018).
It extends research on representation learning for causal inference (Veitch et al., 2019; 2020), but could better position itself in this literature. Connections to theoretical work on ILTs and non-identifiability (Dai et al., 2022) are relevant but lack empirical grounding.

**Theoretical Claims:**

Yes, I check the main proofs.

---

> ### Author Rebuttal · Authors · 2025-04-01
>
> **Additional experiments** can be found at https://anonymous.4open.science/r/icml2025-6599/add_exp_r1.pdf
>
> - - - -
>
> ### **Impact of ILTs on ATE**
>
> > How does ILT invariance specifically impact DML estimation?
>
> The ILT invariance of pre-trained representation does not only have theoretical but also crucial practical consequences in DML estimation. More specifically, Random Forest (RF) and the Lasso are commonly used for nuisance function estimation in DML applications with tabular data. However, as we show both theoretically and empirically, additivity and sparsity cannot reasonably be assumed given the ILT invariance.  Further extending our previous results on the IMDb and X-Ray datasets, we show in Fig.13 that both the ILT non-invariant nuisance estimators RF (building on additivity) and Lasso (building on sparsity) yield biased ATE estimation when using pre-trained representations, while DML with ILT invariant “Linear” (NN with linear and classification model head) estimators yields unbiased results in both empirical studies.
>
> We thank the reviewer for this question and hope our explanation clarifies it. We have discussed this aspect in the paper on pages 7-8, but are happy to further highlight this point in a revised version of the manuscript.
>
> - - - -
>
> ### **With/out Pre-Training**
>
> > Why is there no experiment comparing DML with and without pre-trained representations?
>
> We thank the reviewer for bringing up this important aspect. To validate the benefits of pre-training in our context, we have now conducted experiments where we fit DML with pre-trained representations and compare it with DML without pre-training. As models have to be trained from scratch, we expect pre-training to have less bias than training from scratch.  We demonstrate this on the pneumonia (X-ray) data set, where we used 500 and all 3769 X-rays for ATE estimation and fitted CNNs as nuisance function estimates (both for the propensity score and outcome regression). The results are depicted in Fig.14 and demonstrate that DML using pre-trained representations yields unbiased estimates with good coverage while DML with from-scratch training of CNNs does not. Note that these results become even more pronounced in case of smaller sample sizes (<200) –- a setup that is frequently encountered in clinical practice.
>
> We thank the reviewer again for this very helpful comment. We think that the new results further strengthen our paper.
>
> - - - -
>
> ### **HCM**
>
> > Are there scenarios where HCM does not hold? If real-world data does not follow an HCM structure, how does this affect ATE estimation?
>
> We thank the reviewer for raising this question. Indeed, the HCM structure is a structural assumption that does not necessarily need to hold in all real-world data applications. However, the success of DNNs in many real-world applications and the fact that DNNs precisely mimic such HCM structures suggest that it may be a reasonable assumption in many settings. That being said, our asymptotic normality results in Thm. 5.7 do not necessarily depend on the HCM assumption. In fact, we used it to relax the smoothness assumptions that otherwise would be required. Given a sufficient amount of smoothness of the target function and low ID of the representations, Thm. 5.7 can achieve the same root-n consistency and asymptotic normality. However, there are cases where ATE estimation yields biased results if the HCM structure does not hold. To illustrate this, we conducted an additional experiment, where confounding is based on the product of pre-trained representations (hence HCM no longer holds, which is crucially different from our previous complex conf. experiments). The results in Fig. 15 demonstrate that none of the estimators, not even the DML with NN, can yield unbiased estimates in this non-HCM setup.
>
> We appreciate the reviewer’s thoughtful comment and hope our response and additional experiments clarified the question.
>
> - - - -
>
> ### **Other Comments**
>
> > [The paper] could better position itself in the literature
>
> We thank the reviewer for the comment. We will revise our related literature section accordingly.
>
> - - - -
>
> We sincerely appreciate the reviewer's constructive feedback and hope that the additional experimental findings and clarifying explanations address the points raised in the review. Should the reviewer find the response satisfactory, we would appreciate reconsidering the initial score. Otherwise, we remain fully committed to addressing any remaining concerns during the second author response phase.

---

> > ### Comment · Reviewer_vZqT · 2025-04-05
> >
> > Thanks for the author's reply, I'll raise the score from 2 to 3.

---

> > > ### Author Response · Authors · 2025-04-06
> > >
> > > We sincerely thank the reviewer for their careful assessment, the revised score, and the constructive feedback that helped improve our paper.

---

### Decision · Program_Chairs · 2025-05-01

**Decision:**

Accept (poster)

**Comment:**

This paper presents a theoretically motivated approach to ATE estimation using latent features extracted from pre-trained neural networks, with an emphasis on addressing non-tabular confounding in settings such as image-based data.

The reviewers generally appreciate the depth and rigor of the theoretical contributions. One reviewer highlights the novelty in bridging DML and high-dimensional feature representations, while another notes the originality in adapting convergence rate analysis to neural estimators under nonstandard assumptions.

However, concerns remain, especially regarding the experimental component. Several reviewers point out that the empirical evaluation is limited to toy datasets and lacks critical ablation studies, such as comparisons between DML using raw data vs. latent features. Others question the practical relevance of HCM and ILTs in real-world data, noting that these assumptions, though theoretically sound, are difficult to validate empirically. Moreover, the absence of meaningful empirical support for claims about the robustness of neural network-based ATE estimation weakens the practical implications. Overall, I recommend accepting if there is room.